# In situ modulating coordination fields of single-atom cobalt catalyst for enhanced oxygen reduction reaction

Meihuan Liu[1,2,4], Jing Zhang[1,4], Hui Su ⓘ[3] ✉, Yaling Jiang[1], Wanlin Zhou[1], Chenyu Yang ⓘ[1], Shuowen Bo[1], Jun Pan ⓘ[2] ✉ & Qinghua Liu ⓘ[1] ✉

Single-atom catalysts, especially those with metal−$N_4$ moieties, hold great promise for facilitating the oxygen reduction reaction. However, the symmetrical distribution of electrons within the metal−$N_4$ moiety results in unsatisfactory adsorption strength of intermediates, thereby limiting their performance improvements. Herein, we present atomically coordination-regulated Co single-atom catalysts that comprise a symmetry-broken Cl−Co−$N_4$ moiety, which serves to break the symmetrical electron distribution. In situ characterizations reveal the dynamic evolution of the symmetry-broken Cl−Co−$N_4$ moiety into a coordination-reduced Cl−Co−$N_2$ structure, effectively optimizing the $3d$ electron filling of Co sites toward a reduced $d$-band electron occupancy ($d^{5.8} \rightarrow d^{5.28}$) under reaction conditions for a fast four-electron oxygen reduction reaction process. As a result, the coordination-regulated Co single-atom catalysts deliver a large half-potential of 0.93 V and a mass activity of 5480 A $g_{metal}^{-1}$. Importantly, a Zn-air battery using the coordination-regulated Co single-atom catalysts as the cathode also exhibits a large power density and excellent stability.

The electrochemical oxygen reduction reaction (ORR) serves as a pivotal component in emerging energy technologies such as fuel cells and metal-air batteries, presenting a promising and environmentally friendly avenue for modern sustainable energy conversion and storage[1–3]. Unfortunately, the ORR is a multi-step process involving multi-electron and proton coupling that suffers from sluggish kinetics and intricate mechanisms[4–7]. Although platinum-based materials have historically demonstrated high catalytic efficiency in facilitating the ORR, the high cost, scarcity, and poor stability of platinum have hindered their widespread commercial application[8–10]. Consequently, materials aspiring to function as effective ORR electrocatalysts must possess sufficient reactivity for cleaving the O−O bond, an optimized geometric construction to facilitate efficient $O_2$ diffusion and mass transfer, and robust reactive sites for long-term durability[11,12]. Among

these, single-atom catalysts (SACs), especially those housing isolated single-metal sites anchored in a nitrogen-doped carbon matrix (M−N−C), have garnered significant attention for ORR owing to their tunable electronic properties, distinctive geometric configurations, maximal atomic utilization efficiency, and low cost[13–16]. Consequently, the pursuit of developing exemplary M−N−C-based ORR electrocatalysts with high selectivity, sufficient reactivity, and industrial applicability, is increasingly receiving serious attention, yet remains a formidable challenge.

M−N−C materials, especially those featuring symmetrical planar configurations with four nitrogen-coordinated metal sites (M−$N_4$ moieties), have been extensively investigated and have demonstrated favorable catalytic performance[17–19]. Nevertheless, their ORR activity and four-electron (4e⁻) selectivity still fall short when compared to the

[1]National Synchrotron Radiation Laboratory, University of Science and Technology of China, Hefei 230029 Anhui, China. [2]State Key Laboratory for Powder Metallurgy, Central South University, Changsha 410083 Hunan, China. [3]Key Laboratory of Light Energy Conversion Materials of Hunan Province College, College of Chemistry and Chemical Engineering, Hunan Normal University, Changsha 410081 Hunan, China. [4]These authors contributed equally: Meihuan Liu, Jing Zhang. ✉e-mail: suhui@ustc.edu.cn; jun.pan@csu.edu.cn; qhliu@ustc.edu.cn

benchmark Pt/C. This deficiency is attributed to the unfavorable adsorption capacity for oxygen-related intermediates, resulting from the symmetric distribution of electrons within the planar M-N$_4$ moieties[13,20]. Noticeably, the activity and selectivity of SACs are closely related to the electron occupancy of $d$ orbitals, as they engage in $\sigma$-bonding with oxygen-related species[21,22]. Regrettably, for the 3$d$ transition metals, the more 3$d$-band electrons in the symmetrical M–N$_4$ moieties can lead to reduced orbital overlap with O 2$p$ electrons, necessitating higher energy for the formation and evolution of *OOH species[23]. To address this challenge, the modulation of the coordination environment of metal centers using elements with varying atomic radii and electronegativity is considered a promising strategy to break the planar symmetry of electron density[24,25]. Furthermore, the potential-dependent surface rearrangement of electrocatalysts has been predicted in theoretical calculations and observed in experiments, which can in situ optimize the electron distribution of the active sites to accelerate the adsorption and dissociation of reactive species[26–28]. Fallaciously, the stable symmetrical M–N$_4$ moiety impedes the self-optimization of active centers during the reaction, posing an obstacle for improving their ORR activity. Thus, breaking the symmetric electron distribution of single-atom metal centers is impressive for actuating structural self-reconstruction, in situ optimizing the adsorption strength of oxygen species, and subsequently enhancing the four-electron ORR properties.

Herein, we have successfully fabricated coordination-regulated Co sites (CR-Co) confined in N-doped carbon nanosheets through a facile two-step pyrolysis strategy. At the atomic level, both local electric quadrupole transition (1$s$ to 3$d$ transition) and local electric dipole transition (1$s$ to $p$ transition) are achieved by introducing Cl bonding to form the symmetry-broken Cl–Co–N$_4$ moiety in the N-doped carbon matrix (CR-Co/ClNC), thereby evidencing the breakage of symmetric electron distribution. In situ X-ray absorption fine structure (XAFS) spectroscopy reveals that the Cl–Co–N$_4$ moiety dynamically releases two N coordination to form an unsaturated Cl–Co–N$_2$ structure at the early reaction state. Especially, this structural self-rearrangement can in situ optimize the 3$d$ electron filling of Co sites toward a low $d$-band electron occupancy ($d^{5.8} \rightarrow d^{5.28}$) during the reaction. This, in turn, promotes the transformation of the O–O bond from absorbed oxygen-containing species into O* and further evolution into the (Cl–Co–N$_2$)–O* structure. This transformation has been corroborated by in situ synchrotron radiation infrared spectroscopy (SRIR) results, and it subsequently enhances the four-electron ORR properties. The well-designed CR-Co/ClNC catalyst exhibits significantly improved 4e$^-$ ORR selectivity (98%) with a half-wave potential of 0.93 V and a high mass activity of 5480 A g$_{metal}^{-1}$ at 0.85 V. This represents an approximate 47-fold increase compared to that of commercial Pt/C (116 A g$_{metal}^{-1}$). As a proof of concept, the Zn-air battery assembled with the CR-Co/ClNC catalyst delivers a superior power density of 176.6 mW cm$^{-2}$ compared to Pt/C (127.1 mW cm$^{-2}$), which suggests excellent potential for practical application in metal-air batteries.

## Results and discussion

### Morphology and structure characterizations of electrocatalysts

The atomically dispersed cobalt electrocatalyst in which coordination-regulated Co sites were confined within N-doped carbon nanosheets was prepared via a facile two-step pyrolysis strategy. Initially, ZIF-8 was blended with KCl powder and subjected to pyrolysis under a flowing Ar atmosphere to yield nitrogen-doped carbon (NC) nanosheets with accessible active sites. Subsequently, to regulate the Co coordination field, the coordination of metal sites in the Co salt containing Cl was introduced onto the surface of the NC nanosheet by pyrolysis at different temperatures, in which 300 °C was selected to obtain CR-Co/ClNC. In this process, the Co atoms were simultaneously trapped and anchored in NC due to the strong coupling effect with lone-pair electrons of N species. Meanwhile, the Co/NC sample was prepared as a

reference by elevating the pyrolysis temperature to 800 °C to exclude the Cl component[29].

Transmission electron microscopy (TEM) images demonstrate that the obtained CR-Co/ClNC and Co/NC catalysts possess a sheet-like structure (Fig. 1a and Supplementary Figs. 1 and 2) without obvious nanoparticles. High-angle annular dark-field scanning transmission electron microscopy (HAADF-STEM, Fig. 1b) of CR-Co/ClNC confirms the presence of dense and isolated speckled bright dots marked by red circles, signifying atomically dispersed Co atoms anchored on the NC substrate. This can be further proven by the powder X-ray diffraction (XRD) results in Supplementary Fig. 3, which show the absence of diffraction peaks corresponding to Co species. To further clarify the morphological structure of Co sites, elemental mapping images reveal the homogeneous distribution of C, N, Co, and Cl in the entire CR-Co/ClNC architecture (Fig. 1c) and homogeneously dispersed C, N, and Co in the Co/NC sample (Fig. 1d). The N$_2$ physisorption isotherms reveal a slightly greater specific surface area of CR-Co/ClNC and similar pore architecture in comparison with Co/NC (Supplementary Fig. 4). Significantly, the Co loadings for CR-Co/ClNC and Co/NC were 1.68 wt% and 1.74 wt%, respectively, as determined from inductively coupled plasma–optical emission spectrometry (ICP–OES) analysis. These results confirm the uniform dispersion of Co atoms on the surface sites of the NC substrate and the introduction of Cl in CR-Co/ClNC.

Additionally, in the high-resolution Cl 2$p$ X-ray photoelectron spectroscopy (XPS) spectrum (Supplementary Fig. 5) of CR-Co/ClNC, the characteristic peak at 197.7 eV corresponds to the Co−Cl species, while peaks at 199.4 eV (2$p_{3/2}$) and 200.8 eV (2$p_{1/2}$) are attributed to the C-Cl bond[30].The content of Cl in CR-Co/ClNC is determined to be 2.15 wt % by XPS. The N 1$s$ spectra (Supplementary Fig. 6) also show a peak at 399.4 eV, which can be assigned to the Co−N bond. Collectively, the appearance of Co−Cl and Co−N bonds in the XPS spectra of CR-Co/ClNC indicates the coordination of Co atoms with both Cl and N atoms. More importantly, extended X-ray absorption fine structure (EXAFS) spectra were obtained to further identify the coordination environment of cobalt (Fig. 1e). To exclude the high-frequency noise in the high $k$ region and ensure sufficient independent free points during EXAFS fitting, a $k$ range of 2.4–10.2 Å$^{-1}$ was selected for all Co samples during the fast Fourier transform (Supplementary Fig. 7). In contrast to the Co foil and CoOOH references, the prominent peak located at ~1.5 Å can be assigned to the Co–N/Cl bond, owing to the presence of N and Cl elements, with no Co−Co scattering peak appeared in both CR-Co/ClNC and Co/NC, validating an atomic dispersion of Co in the NC support[31,32]. Following the introduction of Cl into CR-Co/ClNC, the Co −N/Cl peak exhibits a positive shift of 0.15 Å compared to the Co/NC sample, suggesting the formation of a Co−Cl coordination structure due to the larger atomic radius of Cl. Furthermore, to elucidate the local coordination structure of Co sites, the fitted curves of the $k^3$-weighted Co $K$-edge EXAFS spectra in Fig. 1e and Supplementary Fig. 8, along with the corresponding data of CR-Co/ClNC and Co/NC listed in Supplementary Table 1, show that the coordination number (CN) of the Co–N bond for Co/NC is 4.1. For CR-Co/ClNC, CN of 3.9 and 1.0 for the Co–N bond and an axial Co–Cl coordination depicted in Fig. 1e are obtained, respectively. To highlight the local structure of the Co metal site and axial Cl coordination, the local structure model is shown in Fig. 1e. The above results imply that the proposed local coordination environment of CR-Co/ClNC is composed of a symmetry-broken Cl−Co −N$_4$ moiety, suggesting the successful regulation of the coordination field of the Co center by introducing a highly electronegative Cl element.

To gain atomic-level insights into the local electronics of Co, soft X-ray absorption near-edge structure spectroscopy (XANES) for N was performed, as shown in Fig. 2a. The peak at ~398.2 eV is attributed to the excitation of 1$s$→2$p$ hybrid orbital of N−C $\pi$*, while the peak at ~408 eV arises from N−C $\sigma$* transitions[33]. Upon the introduction of Co sites in the NC substrate, new peaks at 399.4 eV were

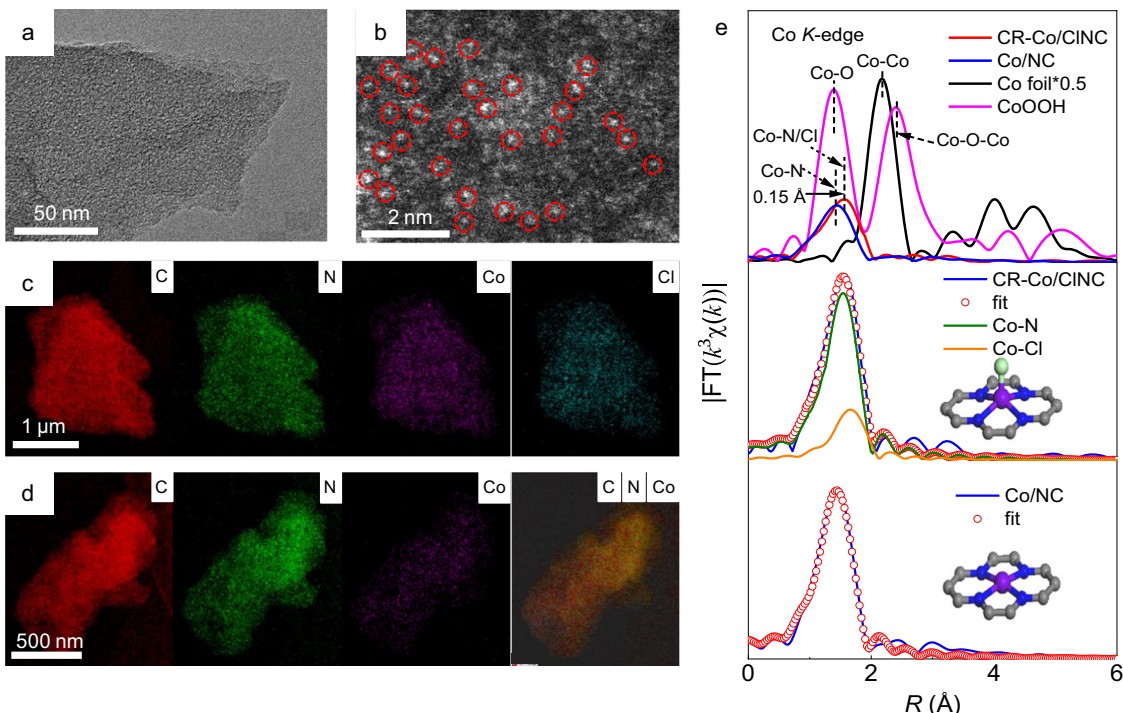

**Fig. 1 | Morphology and structure characterizations. a** TEM and **b** HAADF-STEM images of the CR-Co/ClNC catalyst. TEM-EDS mapping images for **c** CR-Co/ClNC and **d** Co/NC. **e** FT-EXAFS spectra of the Co *K*-edge for the CR-Co/ClNC catalyst and reference samples, and the corresponding fitting curves for CR-Co/ClNC and Co/NC. The inset shows Cl, Co, N, and C atoms indicated by cyan, purple, dark blue, and gray.

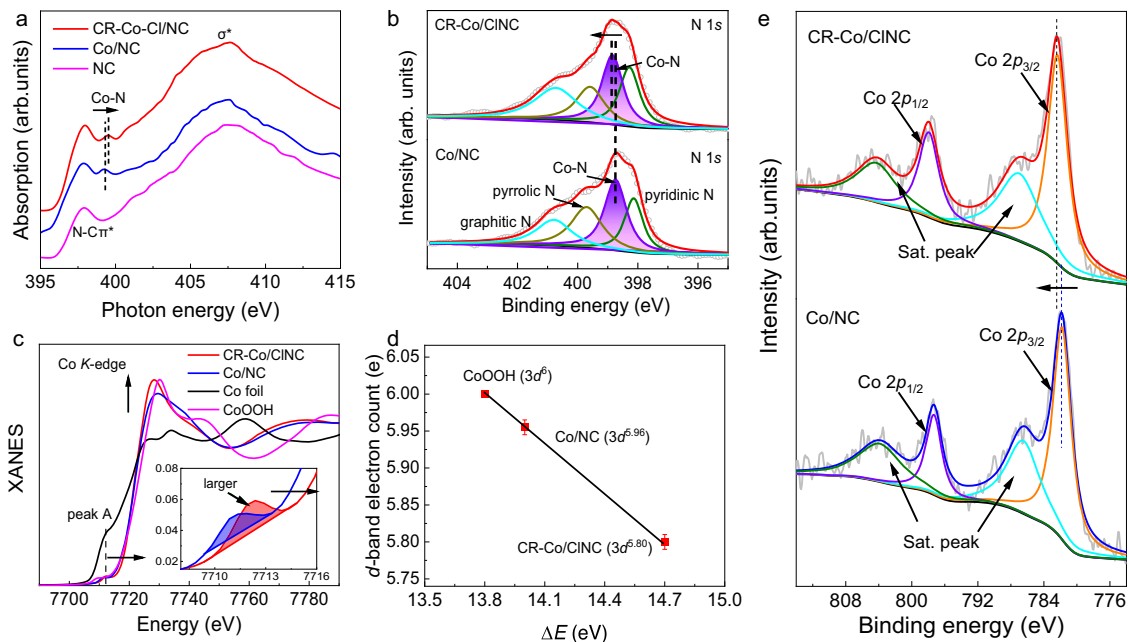

**Fig. 2 | Chemical state and atomic local structure. a** N *K*-edge XANES spectra of CR-Co/ClNC, Co/NC and NC. **b** N 1*s* XPS spectra of CR-Co/ClNC and Co/NC. **c** Co *K*-edge XANES spectra for CR-Co/ClNC and reference samples. The inset shows the fitted pre-edge region. **d** The fitted average formal *d*-band electron counts. The error bars are the standard deviations of three replicate calculation. **e** Co 2*p* XPS spectra of CR-Co/ClNC and Co/NC.

observed for both CR-Co/ClNC and Co/NC, which can be assigned to the excitation of Co−N. For CR-Co/ClNC, the Co−N peak displays a slight positive shift of ~0.2 eV in comparison with Co/NC, suggesting a reduced electron transfer of Co−N after the introduction of Cl elements[26]. Furthermore, the high-resolution XPS N 1*s* spectra of both catalysts can be deconvoluted into pyridinic N, Co−N, pyrrolic N, and graphitic N species (Fig. 2b)[34]. The corresponding ratios of the

deconvoluted N species are summarized in Supplementary Fig. 9. It reveals the presence of N types in similar proportions, thus excluding the influence of the N type in the substrate on ORR performance. Meanwhile, the high-energy shift of binding energy (~0.2 eV) for Co−N in CR-Co/ClNC proves that the reduced interaction between Co and N is attributed to the introduced Cl atoms, which matches well with the N *K*-edge XANES results[35].

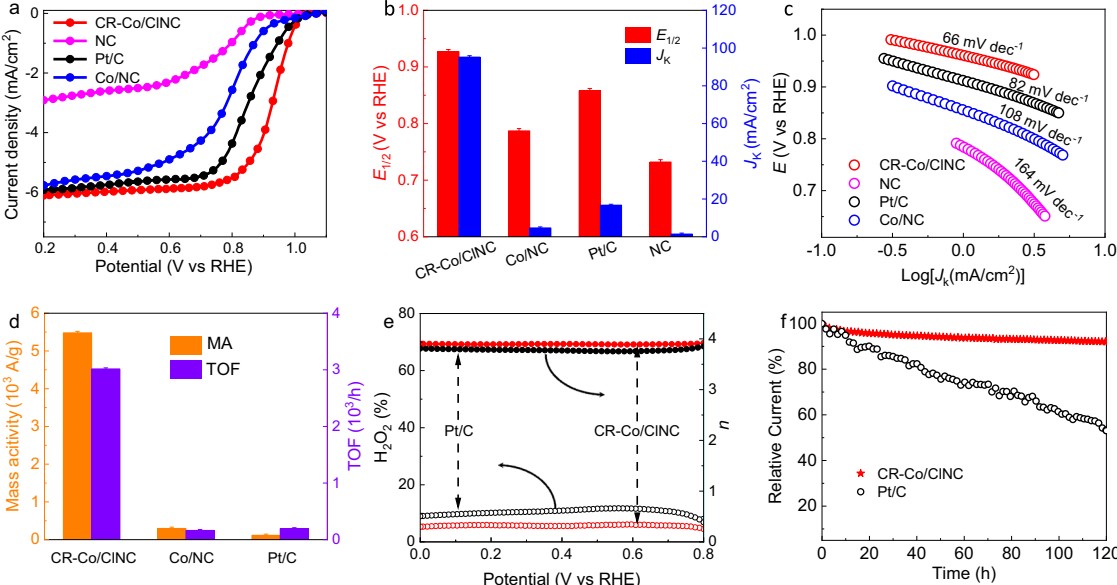

**Fig. 3 | Electrochemical oxygen reduction performance. a** Polarization curves for CR-Co/ClNC and references under 0.1 M $O_2$-saturated KOH, 1600 rpm, scan rate of 10 mV s$^{-1}$. **b** The contrast between CR-Co/ClNC and references for $J_k$ and $E_{1/2}$. The error bars are the standard deviations of three replicate calculation. **c** Tafel slopes and **d** Mass activity (MA) and turnover frequency (TOF) for CR-Co/ClNC and reference samples. The error bars error bars are the standard deviations of three replicate calculation. **e** Electron transfer number (top) and $H_2O_2$ yield (bottom) vs potential of CR-Co/ClNC and Pt/C. **f** Current-time chronoamperometric responses of CR-Co/ClNC and Pt/C.

To obtain in-depth insight into the local electronic structure changes of Co sites after introducing Cl at the atomic scale, XANES spectra of the Co $K$-edge were adopted (Fig. 2c). The XANES spectrum of CR-Co/ClNC display different peak shapes and intensities, indicating a different coordination environment and local electronic structure as compared to Co/NC. The enlarged pre-edge feature (peak A) of CR-Co/ClNC (inset, magnified image in Fig. 2c) suggests the presence of both local electric quadrupole transition ($1s$ to $3d$ transition) and local electric dipole transition ($1s$ to $p$ transition) in the catalyst, clearly confirming a noncentrosymmetric electron distribution of Co sites in CR-Co/ClNC[24]. The maximum value of the first derivative of CR-Co/ClNC exceeds that of CoOOH, revealing a higher Co oxidation state in CR-Co/ClNC related to CoOOH (Supplementary Fig. 10). Additionally, the high-energy shift of the absorption edge and the enhanced intensity of the white-line peak in CR-Co/ClNC demonstrate an increased valence state of Co following the introduction of Cl[36]. The valence state can be obtained by linear fitting of the absorption edge position (Supplementary Fig. 11). Figure 2d illustrates the absorption edge positions of the catalysts as a function of the formal $d$-band electron count obtained from the CoOOH ($3d^6$) standard. Notably, CR-Co/ClNC exhibits a distinct lower number of $d$-band electrons (5.80) compared with Co/NC (5.96), suggesting a pronounced hybridization of Co $3d$–Cl $2p$ states, effectively breaking the symmetric electron distribution of Co−N moieties. The Co $2p_{3/2}$ peak in CR-Co/ClNC shifts to a higher binding energy than that of Co/NC (Fig. 2e), indicating a higher valence state of Co following Cl regulation, which is consistent with the previous XANES results. Consequently, these findings demonstrate that the symmetry-broken Cl−Co−$N_4$ moieties in CR-Co/ClNC effectively break the symmetric electron distribution of Co sites in a typical Co−$N_4$ moiety, leading to a reduced number of $d$-band electrons, and ultimately optimizing the performance of the active sites.

## Electrochemical oxygen reduction performance

To assess the ORR performances, the as-prepared samples and commercial Pt/C were subjected to evaluation using a rotating disk electrode (RDE) in $O_2$-saturated 0.1 M KOH electrolyte. The comparison of linear scan voltammogram (LSV) curves for CR-Co/ClNC, Co/NC, NC, and Pt/C catalysts are illustrated in Fig. 3a. The CR-Co/ClNC exhibits an optimal ORR activity with a half-wave potential ($E_{1/2}$) of 0.93 V vs a reversible hydrogen electrode (RHE, the potentials mentioned below are all relative to RHE) (Fig. 3b), obviously outperforming those of commercial Pt/C, Co/NC and NC. A higher onset potential of 1.008 V is required for CR-Co/ClNC to achieve 5% of the diffusion-limited current (Supplementary Fig. 12), slightly surpassing that of Pt/C (0.994 V). Furthermore, the mass transfer limiting current density for CR-Co/ClNC reaches 6.08 mA cm$^{-2}$, exceeding that observed for the commercial Pt/C catalyst (5.92 mA cm$^{-2}$). This higher limiting current density seems to be a consequence of the fast desorption rates of product on the CR-Co/ClNC electrode. In addition, CR-Co/ClNC possesses a superior kinetic current density ($J_k$) up to 95.2 mA cm$^{-2}$ at 0.85 V, 5.7-fold that of Pt/C (16.7 mA cm$^{-2}$). In contrast, Co/NC displays inferior ORR activity with reduced $J_k$ (4.7 mA cm$^{-2}$) and a more negative $E_{1/2}$ (0.79 V), demonstrating the optimal intrinsic ORR activity of CR-Co/ClNC attributed to suitable modulation of the coordination field of Co sites with Cl elements. Meanwhile, the ORR performance achieved by CR-Co/ClNC is superior to those of most recently reported SACs (Supplementary Table 2). The faster ORR kinetics for CR-Co/ClNC are further confirmed by its smallest Tafel slope (66 mV dec$^{-1}$) compared to Pt/C (82 mV dec$^{-1}$), Co/NC (108 mV dec$^{-1}$), and NC (164 mV dec$^{-1}$) (Fig. 3c). Additionally, as shown in Fig. 3d, CR-Co/ClNC delivers a high turnover frequency (TOF) of 3016 h$^{-1}$ and a large mass activity (MA) of 5480 A g$_{metal}^{-1}$ at 0.85 V, almost 15 and 47 times higher than those of commercial Pt/C (197 h$^{-1}$ and 116 A g$_{metal}^{-1}$). To eliminate the influence of test errors, including the loading of catalyst, the uniformity of electrode film, the flatness of the surface of glassy carbon electrode, and the penetration of electrolyte on the electrode surface, CR-Co/ClNC and Co/NC were tested three times in the repetitive process to obtain the error bars of ECSA (Supplementary Figs. 13–15). Moreover, similar electrochemically active surface area (ECSA) for CR-Co/ClNC (112.4 m$^2$ g$^{-1}$ with an error bar of 7.01 m$^2$ g$^{-1}$) and Co/NC (100.1 m$^2$ g$^{-1}$ with an error bar of 6.21 m$^2$ g$^{-1}$) indicates that the source of enhanced activity of CR-Co/ClNC is the Cl−Co−$N_4$ structure formed by axial Cl coordination, which optimizes the electronic structure to achieve increased intrinsic activity of the active site, rather than the

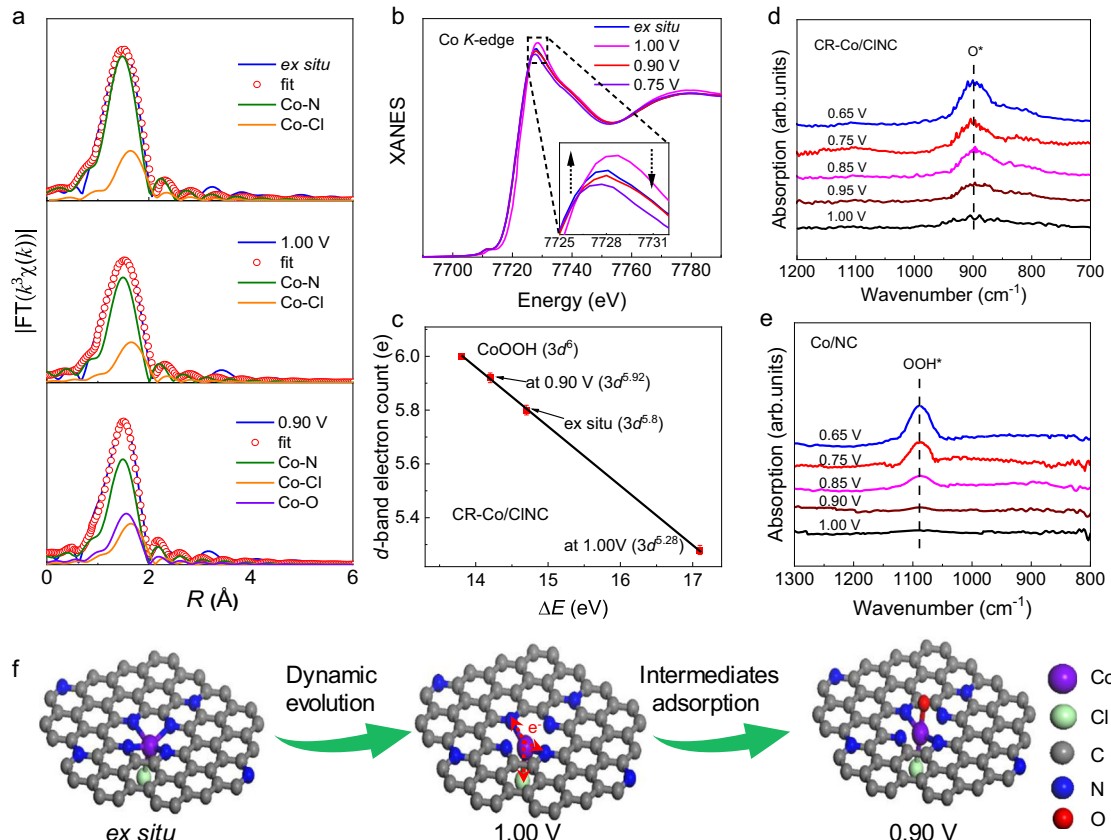

**Fig. 4 | In situ XAFS and SRIR characterizations. a** The curve-fitting analysis of EXAFS spectra and **b** XANES spectra of Co *K*-edge recorded at different applied potentials during the ORR process for CR-Co/ClNC. Inset, magnified white-line peak region. **c** The fitted average formal *d*-band electron counts of Co at CR-Co/ClNC under ex situ, 1.00 V, and 0.90 V conditions based on the absorption edge of Co *K*-edge XANES spectra. The error bars are the standard deviations of three replicate calculation. In situ SRIR measurements under various potentials for (**d**) CR-Co/ClNC and (**e**) Co/NC. **f** ORR schematics of CR-Co/ClNC.

increase in the number of active sites (Supplementary Figs. 16–18). These results demonstrate that the introduction of axial Cl atoms in the symmetry-broken Cl−Co−N₄ moieties with low 3*d*-band electron filling can significantly enhance the ORR activity.

Apart from the ORR catalytic activity, catalytic selectivity is an important index to evaluate the electrocatalytic oxygen reduction performance. As illustrated in Fig. 3e, the electron transfer number (*n*) of CR-Co/ClNC was calculated to be ~3.96 according to the rotating ring disk electrode (RRDE) measurements, confirming the preference of the CR-Co/ClNC catalyst for the 4e⁻ pathway. Moreover, the 4e⁻ ORR pathway is further proven by the Koutechy−Levich (K-L) equation applied in the diffusion-controlled region at various rotation speeds (Supplementary Figs. 19 and 20). Clearly, the peroxide yield for CR-Co/ClNC in the same potential range is below 3%, lower than those of the other samples (Supplementary Fig. 21), indicating the highest four-electron selectivity toward the ORR pathway for CR-Co/ClNC. Additionally, the stability and durability of the CR-Co/ClNC catalyst were measured via chronoamperometry (CA) and accelerated durability testing (ADT). The CR-Co/ClNC exhibits reliable stability with better methanol resistance and lower current density attenuation (<8%) after 120 h of CA testing at 0.7 V, and the CR-Co/ClNC catalyst only experiences a 20 mV loss in $E_{1/2}$ after the ADT, suggesting its long-term durability for the ORR (Fig. 3f and Supplementary Figs. 22 and 23). The morphology and electronic characterizations, such as HAADF-STEM and XAFS of CR-Co/ClNC (Supplementary Figs. 24 and 25) after long-term electrolysis, were performed without observation of Co species and nanoparticles. These results indicate the superior structural robustness and excellent stability of CR-Co/ClNC.

## In situ characterizations of the evolution of active sites
To deeply elucidate the underlying mechanisms responsible for the exceptional four-electron selectivity of CR-Co/ClNC during the ORR process, in situ XAFS measurements of the Co *K*-edge were conducted using a homemade in situ cell (Supplementary Fig. 26a)[37,38]. Firstly, the evolution of the local coordination environment of the Co sites under applied voltages was clarified by in situ EXAFS results (Supplementary Fig. 26). It is evident that a dominant peak at ~1.56 Å showed a 20% damping in peak intensity and a positive shift of 0.04 Å for CR-Co/ClNC as the potential changed from ex situ (immersed in solution without applied voltage) to 1.00 V conditions. The reduced peak intensity in EXAFS clearly signifies the potential-driven reduction in the coordination of Co single sites and no adsorption by oxygen-containing species at the early reaction state. The peak displays a discernible increase in intensity and a slight negative shift in location when applying voltages of 0.90 and 0.75 V, implying the adsorption of oxygen-containing species during the ORR process. Quantitatively, the EXAFS fitting results in Fig. 4a, Supplementary Figs. 27–30, and Supplementary Table 3 exhibit a coordination number of four Co−N bonds and one axial Co−Cl bond under ex situ states, resembling the EXAFS result measured under air conditions. Interestingly, when applying a potential of 1.00 V, the coordination number of the Co−N bonds is evidently reduced to two, indicating that the potential-driven structural evolution of Co single sites truly occurs under ORR conditions by releasing Co centers from the N−C substrate to form a Cl−Co−N₂ active site. It is of high interest that this coordination-unsaturated Cl−Co−N₂ structure favors the surface adsorption of oxygen molecules. As the potential continually decreases to 0.90 V, the Cl−Co−N₂ coordination combined with an additional Co−O coordination is retained for the Co

sites, which might be driven from the adsorption of key reactive oxygen-containing intermediates.

The dynamic evolution of the local coordination structure often accompanies the optimization of the electronic structure of the active site under working conditions. XANES analysis is employed to clarify the changes in the electronic structure. In situ XANES spectra of the Co $K$-edge for CR-Co/ClNC at different applied voltages are depicted in Fig. 4b. Compared with the ex situ conditions, the absorption edge shows a slight positive-energy shift, and the white-line peak displays a modest increase in intensity at an applied voltage of 1.0 V. To quantify the change in $d$-band electron filling, the absorption edge shifts are correlated with the $d$-band electron counts of Co using CoOOH ($3d^6$) as a standard (Fig. 4c and Supplementary Fig. 31). A lower Co $3d$ electron filling count in CR-Co/ClNC (5.28) is observed at 1.0 V. The dynamically evolved Cl−Co−N$_2$ experiences a rapid decrease of 0.52 electrons at the early reaction state (corresponding to a dynamic $3d$ electron evolution of $d^{5.8} \rightarrow d^{5.28}$), implying that there are more $d$-band vacant orbitals for coupling with O $2p$ orbitals to regulate the adsorption of oxygen-containing species. Notably, the Co $3d$ electron filling count in CR-Co/ClNC depopulates much more rapidly and violently with the applied potential than that of Co/NC (Supplementary Fig. 32), which is beneficial for optimizing the adsorption kinetics of intermediates. These results reveal that a fast in situ modulation of $d$-band electrons occurs at the early reaction state for CR-Co/ClNC, as the symmetry-broken Cl−Co−N$_4$ moiety rapidly evolves into the coordination-reduced Cl−Co−N$_2$.

To investigate the adsorption properties of key intermediate species over the Co sites with dynamic $3d$ electron evolution, the surface-sensitive in situ synchrotron SRIR technique was adopted by a dedicated cell[39,40]. As shown in Fig. 4d, when the applied potential is less than 1.00 V for CR-Co/ClNC, the new absorption band at 895 cm$^{-1}$ shows a potential-dependent behavior. This can be attributed to the accumulation of crucial intermediates *O over the Cl−Co−N$_2$ moieties under ORR conditions because the stretching vibration of the oxygen species (*O) is usually in the range of 800−900 cm$^{-1}$ [41]. To further verify the effect of the coordinated Cl bond, in situ SRIR signals of Co/NC were also acquired under the same typical potentials for comparison (Fig. 4e). Noticeably, a new absorption band at 1080 cm$^{-1}$ is observed as the applied voltage decreases. Since the infrared vibration peaks of the *OOH species usually appear in the region of 1000−1100 cm$^{-1}$, the absorption band at 1080 cm$^{-1}$ can be assigned to *OOH[42,43]. Only the absorption band of *O species observed for CR-Co/ClNC during the ORR process clearly implies the rapid cleavage of the O−O bond in the *O−OH intermediate on the dynamically evolved Cl−Co−N$_2$ sites. Therefore, the adsorption strength of the key *OOH is effectively optimized to rapidly evolve into *O species for CR-Co/ClNC owing to the dynamic $3d$ electron evolution ($d^{5.8} \rightarrow d^{5.28}$) at the early reaction state, effectively promoting the four-electron reaction kinetics and enhancing ORR activity and selectivity.

Above all, the dynamic evolution of active sites under working conditions is schematically shown in Fig. 4f. Firstly, the Co centers are dynamically released from the N−C substrate to form a coordination-reduced Cl−Co−N$_2$ active structure at the early reaction state (at 1.0 V conditions), revealing a lower $d$-band electron filling. Remarkably, these coordination-reduced Cl−Co−N$_2$ moieties obviously optimize the adsorption of oxygen-containing intermediates in situ and promote the cleavage of the O−O bond for fast reaction kinetics. Finally, the (Cl−Co−N$_2$) −*O active structure is retained as the potential steadily decreases to 0.75 V, thereby realizing a highly efficient ORR process. Moreover, the dynamically evolved electron and coordination structures of the Co sites return to their original state after the reaction according to the XAFS results (Supplementary Fig. 33), suggesting that the structural evolution of the Co active site is a dynamic reversible process. These results suggest that the in situ modulated $d$-band electron filling of Co sites through the introduction of Cl can yield

significantly accelerated ORR kinetics, which endows the CR-Co/ClNC catalyst with excellent potential for industrial applications.

## Zn-air battery (ZAB) performance

In light of the superior ORR performance, the as-prepared CR-Co/ClNC was served as a cathode catalyst in aqueous primary zinc-air batteries (ZABs) to investigate its practicability (Supplementary Fig. 34). The same device was assembled using commercial Pt/C for comparison. The CR-Co/ClNC-incorporated Zn-air cell exhibits an impressive open-circuit voltage of 1.50 V (Fig. 5a), which is higher than that of the Pt/C-based Zn-air cell (1.45 V), indicating a higher output voltage for the battery when using CR-Co/ClNC as the cathode catalyst. From the discharge polarization curves and power density plots for the Zn-air battery (Fig. 5b), it can be seen that the CR-Co/ClNC-based ZAB presents a higher discharging voltage plateau with a maximum power density close to 176.6 mW cm$^{-2}$, which outperforms the Pt/C counterpart (127.1 mW cm$^{-2}$). The power density of CR-Co/ClNC based ZAB surpasses most of the reported SACs (Supplementary Table 4). Notably, the ZAB with CR-Co/ClNC delivers a specific capacity of 745 mA h g$_{Zn}^{-1}$ at a discharging current density of 10 mA cm$^{-2}$ in Fig. 5c, outperforming that of Pt /C-based ZAB (719 mA h g$_{Zn}^{-1}$). The galvanostatic discharge observations at current densities ranging from 5 to 50 mA cm$^{-2}$ are shown in Fig. 5d. The CR-Co/ClNC-based ZAB obviously maintains higher discharge voltages than Pt/C-based ZAB under the current density ranges and then periodically return to 5 mA cm$^{-2}$. Following a high-rate discharge at 50 mA cm$^{-2}$, the discharge potential for the CR-Co/ClNC-based ZAB recovers to the starting stage (5 mA cm$^{-2}$), indicating the good rate capability and reversibility of the CR-Co/ClNC-based ZAB, which can be attributed to the efficient ORR activity and good stability of the CR-Co/ClNC catalyst. Noticeably, no discernible degradation was recorded after 3 cycles for ZAB with the CR-Co/ClNC electrocatalyst over a duration of 30 h at 10 mA cm$^{-2}$ (Supplementary Fig. 35). In addition, the galvanostatic discharge −charge measurements show negligible deterioration of the discharge −charge voltage, reflecting the promising durability of the CR-Co/ClNC in Zn-air batteries (Supplementary Fig. 36). These results suggest that the CR-Co/ClNC catalyst holds good potential for application in ZABs.

In summary, we have constructed a kind of atomically dispersed and coordination-regulated CR-Co/ClNC catalyst with symmetry-broken Cl−Co−N$_4$ moieties via a facile two-step pyrolysis strategy, successfully avoiding the introduction of impurities and realizing the coupling of axial Cl over the Co−N$_4$ moiety. At the atomic scale, we have demonstrated that the introduction of axial Cl atoms in the active site could effectively break the symmetric electron distribution of Co −N$_4$ moieties, achieving low $3d$-band electron filling for in situ regulation of the adsorption strength of oxygen adsorbates around the Co sites. Mechanistic studies employing in situ SRIR and XAFS spectroscopies have revealed the dynamic evolution of the coordination-reduced Cl−Co−N$_2$ structure could optimize the $3d$-band electron occupancy of Co sites in situ at the early reaction state, which is quite beneficial for the cleavage of the O−O (*OOH species) into *O intermediates toward excellent electrocatalytic ORR activity and selectivity. The well-defined CR-Co/ClNC catalyst delivers an appreciable four-electron selectivity and a high kinetic current density exceeding those of Pt/C. Furthermore, the ZAB with the CR-Co/ClNC catalyst presents high efficiency and robust stability at a high current density. This work highlights the importance of modulating the electronic structure and deepens the understanding of the dynamic evolution of single-atom catalysts in promoting ORR performance.

## Methods
### Synthesis of ZIF-8
In a typical procedure, 2-methylimidazole (14.575 g) was dissolved in methanol (250 mL). Subsequently, methanol (250 mL) containing Zn(NO$_3$)$_2$•6H$_2$O (6.375 g) was added to the above solution under

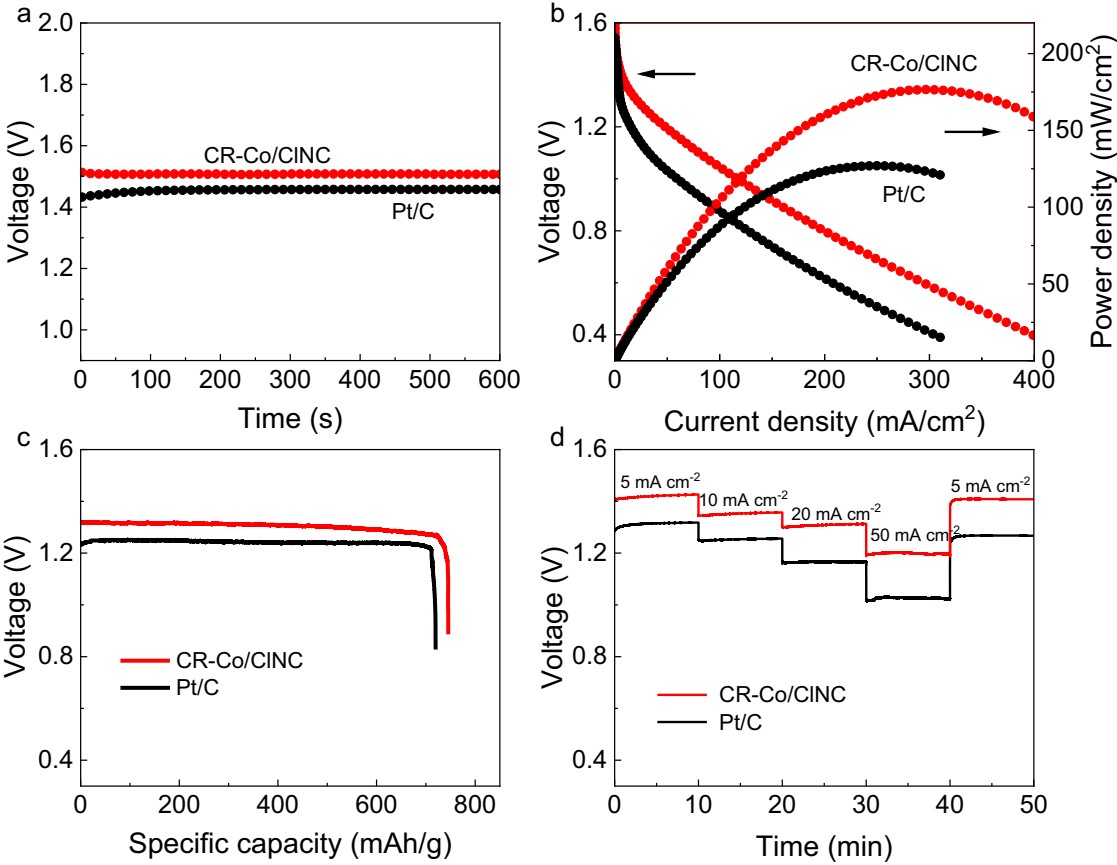

**Fig. 5 | Zn-air battery (ZAB) performance. a** Open-circuit voltage plots of Zn-air batteries assembled with CR-Co/ClNC and Pt/C. **b** Discharge polarization curves and power density plots of CR-Co/ClNC and Pt/C-based Zn−air batteries. **c** Specific capacity by mass normalization at a current density of 10 mA cm⁻². **d** Galvanostatic discharge curves under different current densities.

continuous stirring for 120 min at room temperature. After that, the above milky white dispersion was kept at room temperature for 12 h without stirring. The resultant sample was then centrifuged at 10,732 × *g*, washed with methanol three times and further dried under vacuum for 12 h at 80 °C.

### Synthesis of NC nanosheets
The ZIF-8 (2.5 g) obtained as described above and 10 g KCl were mixed together and ground thoroughly, followed by dissolution in 200 mL of water. After ultrasonication for 2 h, the aqueous solution in the sample was removed by direct centrifugation and dried at 80 °C overnight. The obtained powder was fully ground and heated to 800 °C (2 °C min⁻¹) in an Ar atmosphere for 3 h. Subsequently, the obtained sample was fully ground, stirred with 2 mol/L HCl for 2 h, washed with 2 mol/L HCl three times, then washed with water and ethanol twice each, and finally dried overnight under vacuum at 80 °C.

### Synthesis of the CR-Co/ClNC catalyst
The NC (100 mg) was first dispersed into a 20 mL ethanol solution and subjected to ultrasonication for 10 min. After uniform dispersion, 20 mg cobalt chloride hexahydrate was added to the above solution and ultrasonicated for 10 min. Then, the solution was sealed and magnetically stirred for 12 h under rotary evaporation at 45 °C. The resulting sample was transferred to a vacuum drying oven at 80 °C when the solution was almost volatilized. The obtained powder was fully ground and heated to 300 °C (5 °C min⁻¹) in an Ar atmosphere for 5 h. Thereafter, the powder was washed three times with a mixed solution of water and ethanol, centrifuged after ultrasonication and dried under vacuum at 80 °C.

### Synthesis of Co/NC catalyst
The Co/NC sample was prepared following the same procedure of CR-Co/ClNC but annealed at 800 °C (5 °C min⁻¹).

### Morphology and structure characterization
Powder X-ray diffraction (XRD) measurements were acquired on a Philips X' Pert Pro Super X-ray diffractometer with Cu *K*α radiation. X-ray photoelectron spectroscopy (XPS) was recorded on an ESCALAB MKII with Mg *K*α (hυ = 1253.6 eV) as the excitation source. Scanning electron microscopy (SEM) analyses were carried out on a scanning electron microscope (Gemini SEM 500). Transmission electron microscopy (TEM) and scanning transmission electron microscopy-energy dispersive spectroscopy (STEM-EDS) were conducted using a microscope (JEM−2100F, at 200 kV), and aberration corrector high-angle annular dark-field transmission electron microscopy (AC-HAADF-TEM, JEM-ARM200F) was performed at 200 kV with a probe spherical aberration corrector.

### Electrochemical characterization
Catalyst ink was prepared by sonicating the mixture comprising the catalyst (5 mg), Nafion solution (5 wt.%, 30 µL), and solvent (1000 µL, water/ethanol = 1:3, v/v) for several hours until a homogeneous suspension was formed. Afterward, a certain amount of the catalyst ink (5 µL) was coated and dried on the surface of the polished glassy carbon electrode (3 mm) at room temperature, and the mass of catalyst deposited on the electrode is 3.57 g m⁻². All electrochemical measurements were performed on a CHI760E electrochemical workstation (CH Instruments, China) with a standard three-electrode system using a carbon rod, an Ag/AgCl (saturated KCl) electrode, and

a glassy carbon electrode coated with catalyst serving as the counter, reference, and working electrodes, respectively. Polarization curves for the ORR were recorded at room temperature in $O_2$-saturated 0.1 M KOH aqueous solution at various rotation rates (400–2500 rpm). For the current-time chronoamperometric test, $O_2$ was bubbled into 0.1 M KOH electrolyte for 30 min prior to the experiment and a flow of $O_2$ was maintained over the electrolyte during the test to ensure oxygen saturation. The test process was constant at 0.7 V. In this work, the final potential was calibrated into a reversible hydrogen electrode (RHE), unless otherwise noted.

The transfer electron number (n) of the ORR was calculated according to the Koutecky-Levich equation below:

$$\frac{1}{J} = \frac{1}{J_L} + \frac{1}{J_k} = \frac{1}{B\omega^{1/2}} + \frac{1}{J_K} \tag{1}$$

$$B = 0.62nFC_0D_0^{2/3}V^{-1/6} \tag{2}$$

where $J$ is the measured current density and $J_K$ and $J_L$ are the charge-transfer kinetics and the diffusion-limited current densities, respectively. $\omega$ stands for the angular velocity of the rotating electrode (rad·s$^{-1}$), $n$ is the electron-transfer number in ORR, $F$ is the Faraday constant (96,485 C·mol$^{-1}$), and $C_O$ is the bulk concentration of oxygen. $D_O$ indicates the diffusion coefficient of oxygen, and $V$ is the kinematic viscosity of the electrolyte. $C_O$, $D_O$ and $V$ in 0.1 M KOH electrolyte were $1.2 \times 10^{-6}$ mol·cm$^{-3}$, $1.9 \times 10^{-5}$ cm$^2$·s$^{-1}$ and 0.01 cm$^2$·s$^{-1}$, respectively.

RRDE measurements were carried out to calculate the yield of hydrogen peroxide ($H_2O_2$%) and the corresponding electron transfer number ($n$) as follows:

$$H_2O_2(\%) = 200 \times \frac{I_r/N}{I_d + I_r/N} \tag{3}$$

$$n = 4 \times \frac{I_d}{I_d + I_r/N} \tag{4}$$

where $I_d$ and $I_r$ are the disk and ring currents, respectively. The Pt ring current collection efficiency $N$ was determined to be 0.37.

### Calculation for mass activity and turnover frequency

The mass activity can be determined by considering the metal sites as the active sites. Initially, the metal content of the CR-Co/ClNC catalyst was calculated from the Co loading mass on the electrode. Afterwards, the mass activity (A g$_{metal}$$^{-1}$) was calculated according to the following equation: the geometric electrode area ($A$) of 0.07 cm$^{-2}$ and the metal loading mass of the electrocatalyst ($M_{metal}$, g) were taken into account.

$$\text{Mass activity} = \frac{A \times J_K \times 10^{-3}}{M_{metal}} \tag{5}$$

Similarly, by assuming each Co atom in the catalyst as an active single-site, the number of active sites in the CR-Co/ClNC catalyst could then be calculated based on the Co loading mass on the electrode. The turnover frequency (TOF) can be derived from the following equation:

$$TOF = \frac{J_K \times A \times Ne \times 10^{-3}}{4 \times Mmetal \times N_A/M} \tag{6}$$

where $J_k$ is the kinetic current density (mA cm$^{-2}$), $Ne$ is the electron number per Coulomb ($6.24 \times 10^{18}$), $M_{metal}$ is the metal loading mass on the electrode, $N_A$ is Avogadro's constant ($6.02 \times 10^{23}$), and $M$ is the molar mass of Co (59 g mol$^{-1}$).

### Assembly and electrochemical testing of Zn-air batteries (ZAB)

The liquid ZAB was evaluated using a cyclically home-made instrument under ambient atmospheric conditions. The electrolyte was composed of 6 M KOH with 0.2 M zinc acetate, and a flow of $O_2$ (20 sccm) was maintained into the electrolyte during the test to ensure $O_2$ saturation. The catalysts coated on the carbon paper were used as the membrane electrode assembly (MEA) of the cathode (catalysts coating area was controlled at 1 cm × 1 cm), and a zinc plate with an effective area of 1 cm × 1 cm served as the anode. The polarization curves were recorded by LSV at room temperature on a CHI 760E electrochemical workstation. Both the current density and power density were normalized to the effective surface area of the air electrode. The specific capacity was calculated according to the following equations:

$$\text{Specific capacity} = I \times t/w_{Zn} \tag{7}$$

where $I$ is the applied current (A), $t$ is the serving time (s), and $w_{Zn}$ stands for the weight of zinc consumed (g).

### In situ XAFS measurements

The in situ XAFS measurements were performed at the 1W1B station in the Beijing Synchrotron Radiation Facility (BSRF), China. The maximum current is 250 mA of the storage ring of BSRF. The line station used a Si (111) double-crystal monochromator, and further detuning of 30% to remove higher harmonics when performing the Co K-edge XAFS measurements. During In situ XAFS measurements, the Co catalyst-coated carbon cloths was selected as working electrode in an alkaline solution by a homemade cell through a three-electrode system. Specifically, the 5 mg sample was evenly dispersed in 1 mL solution (water/ethanol = 1:1, v/v) containing 20 μL Nafion. Then, the catalyst ink was deposited as the working electrode on the carbon paper (-1 cm × 1 cm), with the Capton film fixed on the back of the carbon paper to ensure that more opening sites can participate in ORR. The representative potentials (1.00–0.75 V) were selected to obtain the evolution of atomic and electronic structure of the Co sites during the ORR. When collecting XAFS measurement data, we used a Co foil standard sample to calibrate the position of the absorption edge ($E_0$), and collected all XAFS data over a beam time period. We used a Lytle detector (fluorescence mode) to record XAFS spectra during electrochemical reactions.

### In situ SRIR measurements

We conducted in situ SRIR measurements at the beamline BL01B of the National Synchrotron Radiation Laboratory (NSRL, China). In order to gain better infrared signals, electrochemical tests were performed in a homemade top-plate cell. Specifically, considering the vibration absorption of water molecules, we used a ZnSe crystal window and pressed the catalyst electrode close to the window to reduce the loss of infrared light. The SRIR tests were employed in reflection mode with a spectral resolution of 2 cm$^{-1}$. Systemic ORR measurements were made after the background spectrum of the catalyst electrode was measured at an open-circuit voltage, and the ORR potential ranged from 1.00 to 0.65 V.

## Data availability

The data supporting the findings of this study are available within the article and in Supplementary information files. All data are available from the authors upon request.

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

## Acknowledgements

This work was supported by the National Key R&D Program of China (2022YFA1502903 (Q.L.)), the National Natural Science Foundation of China (12205300 (H.S.), 22241202 (Q.L.), and 12135012 (H.S.)), the Natural Science Foundation of Anhui Province (2208085J01 (Q.L.) and 2208085QA28 (H.S.)), and the Key Program of Research and Development of Hefei Science Center, CAS (2022HSC-KPRD003 (W.Z.)). This work was partially carried out at the Instruments Center for Physical Science, University of Science and Technology of China.

## Author contributions

Q.L., H.S. and J.P. conceived the project. M.L. and J.Z. carried out the experiments. Q.L., H.S., M.L., Y.J., W.Z., C.Y., and S.B. analyzed the experimental data. The manuscript was written by M.L., H.S. and Q.L. with contributions from all authors.

## Competing interests

The authors declare no competing interests.
