## [Peer Review File · Nature Communications]

REVIEWER COMMENTS

Reviewer #1 (Remarks to the Author):

In this manuscript, the preparation of symmetry broken Cl coordinated Co-N₄ single atom catalyst, its ORR activity, and its evolution under ORR conditions are reported. The authors also demonstrated the catalysts fine performance for Zn-air battery during discharging. Although extensive synchrotron-based characterization was conducted to decipher the local coordination structure of the single atom cobalt catalyst, a fair number of phenomena in the presented data lacks explanation and elaboration. The electrochemical testing results should also be carried out more stringently. From the reviewer's point of view, this manuscript is interesting but does not meet the standard to be published in Nature Communications. A more detailed point-to-point comment is listed below:

1. If possible, please conduct N₂ adsorption/desorption isotherms and provide associated information on the surface area and pore size distribution of Co/NC and CR-Co/CINC.
2. Fig. 1e: The fit for CR-Co/CINC does not match the experiment results at low and high R values. Specifically, the fit disagrees at 1-1.2 Å and above 2 Å. The later also holds for the fit of Co/CN.
3. Fig. 3a: There appears to be a huge difference in limiting current density among the various sample tested? The authors should explain the origin of such deviation else the credibility of the electrochemical measurements is questionable. Please also provide reproducibility results. Finally, assuming the ORR polarization curves are performed under 1600 rpm, the limiting current of CR-Co/CINC appears to exceed the limiting current a bit.
4. Based on Fig. S12, the ECSA is very different for all catalysts compared in this study. The authors should try their best to maintain similar level of roughness factor when testing the ORR activity of these materials, as least for CR-Co/CINC and Co/NC.
5. Fig. S13: Why is the limiting current of Pt/C never reached?
6. Fig. 3: The author should specify the testing conditions in the figure captions including essential information like the type of electrolyte, scan rate, additives, etc.
7. Fig. S19: The EXAFS spectra look very different from those presented in the main text and Figure S6, particularly at R values ranging from 2-4 Å
8. Fig. 4a: The fittings here again fail to capture the experimental data at R-values in the range of 1.1-1.2 Å and above 2 Å. The peaks located above 2Å clearly decreased when the potential dropped to 0.9 V while the fits are the same for all three. This might be very important information in longer range coordination. The authors however failed to discuss this part.
9. Fig. 4a: what exactly is the rational of saying Co-O bonds do not exist for the ex-situ and 1.00 V? The ex-situ XANES seems to be quite similar to CoOOH in Fig. 2c.
10. Fig. 4b: Is the change presented here reversible?

11. The authors demonstrate the high performance of the CR-Co/CINC for Zn-air batteries during discharge. However, can this material also survive during charging? The authors should provide such information.

12. The performance of Pt/C in the Zn-air battery is under-rated when compared with other literature. (For example, *Chemical Engineering Journal*, 2023, 143855). The performance of CR-Co/CINC is therefore not that impressive.

Reviewer #2 (Remarks to the Author):

In this manuscript, the authors present an atomic coordination-regulated single-atom cobalt catalyst that comprises of asymmetric Co-ClN₄ moiety, which exhibited excellent electrocatalytic ORR performance compared to the commercial Pt/C catalyst and demonstrated promising application for zinc air battery. The authors did systematic experiments to investigate the structure of as-prepared catalyst and also performed in-situ experiments to study the structural variation during the catalytic reaction. Although the strategy of doping various elements to regulate the coordination of single atom is very common (*Adv. Mater.* 2023, 35, 2302485; *Energy Environ. Sci.* 2023, 16, 2629–2636) and the similar materials have been reported in the literatures (*J. Am. Chem. Soc.* 2022, 144, 14505–14516; *Angew. Chem. Int. Ed.* 2020, 59, 6122–6127), the mechanism of the enhanced activity discovered by the authors is interesting. Overall, the manuscript is logically organized and the experiments are well designed, however, I feel that the novelty of this work needs to be further clarified, and some other questions also need to be addressed before acceptance.

Below are my specific comments:

1. In the Introduction part, the authors describe the design rationale of the single-atom catalysts in detail, i.e., breaking the symmetry of M-N₄ moiety to enhance the four-electron ORR process. As far as I know, there are numerous articles that present the coordination modulation of M-N₄ based single-atom catalysts through heteroatom doping, such as P, O, S, I, Cl doping and so on. Then, what is the new point of this work in terms of catalyst design? And, what are the new findings that can inspire further catalyst design and preparation?
2. The ORR performance was tested in 0.1 M KOH, which is a strong alkaline electrolyte. As known, the reference electrode used for the alkaline electrocatalytic reaction should be Hg/HgO, but the authors use Ag/AgCl. Indeed, the Ag/AgCl electrode is unstable in the alkaline electrolyte, which may cause inaccurate determination of the reaction potential.
3. Can the authors determine the weight ratio of Cl in the CR-Co/CINC catalyst? And, the authors showed C-Cl coordination in Cl 2p XPS, this is not consistent with the structure model used in Figure 1e demonstrating Co-Cl coordination. In addition, in the XPS survey spectrum of Co/NC catalyst that is supposed to be free of Cl atoms, however, a weak Cl 1s peak also appeared (Figure S4a). Is there an explanation for this?
4. For Co K-edge, the authors should compare the adsorption edge to confirm the valence state (*Nat. Commun.* 2022, 13, 7754; *ACS Catal.* 2022, 12, 3138–3148). As depicted in Figure 2c, the adsorption

edges of Co/NC and CR-Co/CINC are located between Co foil (0) and CoOOH (+3). However, in Figure S7, their oxidation states are calculated to be larger than 3.

5. Some experimental details need to be added. For Figure 3f, the author should add how to conduct the CA test. Whether there is the oxygen continuously purged into the solution or not during the stability test? Indeed, it seems more common to use the CV scanning method to conduct the ORR stability test. Furthermore, I am confused about the characterizations in Figure S19, shown that there is almost no change happened on the catalyst after ORR stability test. However, the decreased valences states of CR-Co/CINC with the decreased potentials can be obviously seen in Figure S25. Also, a mistake was made in Figure S25, in which the figures are not corresponded by the figure captions. Additionally, the electrochemical method used for conducting the in-situ XAS test should also be added.

6. The authors clarified that the introduction of Cl can in situ optimize the 3d electron filling of Co sites towards a low d-band electron occupancy ($d_{5.8} \rightarrow d_{5.28}$), which promotes the formation of O^* and enhances 4e- ORR selectivity. To make this conclusion more convincing, I suggest the d-band electron occupancy of Co/NC with symmetric Co-N₄ moiety should be calculated.

7. The phenomena found by the in-situ XAFS and SRIR are interesting, the signals of O^* and OOH^* are obvious in Figures 4d and 4e. Indeed, in the recent literature (J. Am. Chem. Soc. 2022, 144, 14505–14516), they clarified that the pyrrole-type Co-N₄ is mainly responsible for 2e- ORR to generate H₂O₂, while the pyridine-type Co-N₄ can effectively catalyze the 4e- ORR. Therefore, I am wondering if there is any possibility that the Co/NC synthesized in this work is pyrrole-type and the incorporation of Cl will change its nature to the pyridine-type? I recommend the authors should investigate the detailed structures of the supports.

8. The details of ZAB test should also be added, such as the flow rate of the O₂-saturated electrolyte, the effective area of the Zn plate and the working electrode. Actually, the discharge curve was carried out for only 10 min under each current density, so it is not appropriate to describe the CR-Co/CINC possesses a high stability in ZAB. In addition, the figure caption is not correctly describing the Figure S26.

Response to Reviewers' Comments

We are grateful to the reviewers for having given us important and valuable comments on the manuscript NCOMMS-23-41656 entitled: “*In situ* modulated coordination fields of single-atom cobalt at the early reaction state for enhanced oxygen reduction reaction”. The detailed replies to your comments are presented in a point-to-point manner as follows. The modifications in the revised manuscript are highlighted in the yellow background.

Reply to Reviewer #1

In this manuscript, the preparation of symmetry broken Cl coordinated Co-N₄ single atom catalyst, its ORR activity, and its evolution under ORR conditions are reported. The authors also demonstrated the catalysts fine performance for Zn-air battery during discharging. Although extensive synchrotron-based characterization was conducted to decipher the local coordination structure of the single atom cobalt catalyst, a fair number of phenomena in the presented data lacks explanation and elaboration. The electrochemical testing results should also be carried out more stringently. From the reviewer's point of view, this manuscript is interesting but does not meet the standard to be published in Nature Communications. A more detailed point-to-point comment is listed below:

First of all, we greatly appreciate your comments and constructive suggestions on our work. We have seriously considered what the problem existed in this work and carefully revised the manuscript. We hope you could give more thought to this manuscript, and your constructive suggestions are extremely valuable in improving the quality and integrity of this manuscript.

1. Comment: If possible, please conduct N₂ adsorption/desorption isotherms and provide associated information on the surface area and pore size distribution of Co/NC and CR-Co/CINC.

Reply: Thank you for your constructive suggestion, which is quite useful for improving the quality of this work. According to your suggestion, the N₂ adsorption/desorption

isotherms of Co/NC and CR-Co/CINC were performed with the results shown in Fig. N1, and the surface area and pore size distribution of Co/NC and CR-Co/CINC are analyzed in the revised manuscript.

The associated surface area and pore size distribution play an important role in the catalytic activity (*Small* **19**, 2300373 (2023)). The N₂ adsorption/desorption isotherms of Co/NC and CR-Co/CINC were obtained to determine the surface area and pore size distribution, as shown in Fig. N1. The CR-Co/CINC catalyst possesses a slightly higher specific surface area of 832.6 m² g⁻¹ than Co/NC (771.4 m² g⁻¹), indicating a larger number of accessible active sites for CR-Co/CINC. Furthermore, the pore size distribution in Fig. N1b-c shows that the pore sizes of the CR-Co/CINC and Co/NC catalysts are mainly distributed in the range of 0.4–1.6 nm. This reveals that CR-Co/CINC and Co/NC have similar pore architectures. Above all, the accessibility of Co sites and mass transport to active sites for CR-Co/CINC is slightly superior to Co/NC during the ORR process, suggesting that the significantly enhanced ORR performance of CR-Co/CINC is mainly derived from local optimization of electronic and coordination structures.

Fig. N1. (a) Nitrogen adsorption-desorption isotherms of the CR-Co/CINC and Co/NC catalysts. Corresponding pore size distribution of CR-Co/CINC and Co/NC catalysts based on the Barrett-Joyner-Halenda (BJH) method (b) and Horvath-Kawazoe (HK) method (c).

Accordingly, Fig. N1 has been added in the revised Supplementary Information as Supplementary Fig. 4. In line 12, page 5 of the revised manuscript, the following text has been added: “The N₂ physisorption isotherms reveal a slightly higher specific surface area of CR-Co/CINC and similar pore architecture in comparison with Co/NC (Supplementary Fig. 4).”

2. Comment: Fig. 1e: The fit for CR-Co/CINC does not match the experiment results at low and high R values. Specifically, the fit disagrees at 1-1.2 Å and above 2 Å. The later also holds for the fit of Co/CN.

Reply: Thank you for your nice question and careful inspection. We are sorry that the EXAFS fitting results did not match the experimental results at low and high values in the previous manuscript. Attentively, *in situ* XAFS measurements during the ORR for single atom electrocatalysts, the low content of metal elements and the absorption of X-ray by aqueous solution will inevitably cause signal interference in XAFS measurements. In particular, it brings a bad data signal-to-noise ratio in the high k part of Co K -edge XAFS, suggesting that the peak in the range of 2–4 Å may come from the contribution of a bad data signal-to-noise ratio in the high k part. In addition, the ground signal around 1 Å is affected by low frequency noise. To eliminate the interference of a bad data signal-to-noise ratio in the high k part, the k range of 2.4–10.2 Å⁻¹ was selected for the fast Fourier transform, and the EXAFS data were refitted as shown in Fig. N2 according to your suggestion. Moreover, for single-atom catalysts, we mainly studied the coordination structure information of its first-shell coordination, so the curve fitting was done on the k^3 -weighted EXAFS function $\chi(k)$ data in the R -range of 1.0–2.2 Å. The new fitting results in Fig. N2 show coordination numbers (CNs) of 3.9 and 1.0 for the Co–N bond and axial Co–Cl coordination for CR-Co/CINC. At the same time, we find that the peaks in the range of 2–4 Å are indeed from the bad data signal-to-noise ratio in the high k part during the fast Fourier transform, and the refitting curves match well in the range of 1–2 Å in the Co K -edge EXAFS.

Fig. N2. (a) FT-EXAFS spectra of the Co *K*-edge for the CR-Co/CINC catalyst and reference samples, and the corresponding fitting curves for CR-Co/CINC and Co/NC. (b, d) The fitting curve of the k^3 -weighted EXAFS spectrum and (c, e) the $\text{Re}(k^3\chi(k))$ oscillation curve for CR-Co/CINC and Co/NC, respectively.

Accordingly, Fig. 1e has been replaced by Fig. N2a in the revised manuscript, and previous Supplementary Fig. 6 has been replaced by Figs. N2b–e as Supplementary Fig. 8 in the revised Supplementary Information.

3. Comment: Fig. 3a: There appears to be a huge difference in limiting current density among the various sample tested? The authors should explain the origin of such deviation else the credibility of the electrochemical measurements is questionable. Please also provide reproducibility results. Finally, assuming the ORR polarization curves are performed under 1600 rpm, the limiting current of CR-Co/CINC appears to exceed the limiting current a bit.

Reply: We thank the reviewer for this constructive suggestion and really compliment the reviewer for your expert knowledge in the field of electrochemistry. According to your suggestion, the difference in limiting current density among the various samples will be discussed in detail.

In this work, the calculation of electrochemical performance is based on the normalization of the electrode geometric area. In addition, considering that the coarseness will be flattened by the diffusion layer, the total limiting diffusion current should be the same theoretically, but there are some small differences due to the

different desorption rates of the products on different electrode surfaces. The limiting current density, associated with the mass transfer of oxygen reactants, is an important parameter for characterizing the intrinsic activity and kinetic process of the catalytic ORR. With the increase of the electrode rotation speed and reactant species diffusion speed, the current density of the electrocatalyst will increase along with the decrease of the thickness of the diffusion layer. It should be noted that the limiting diffusion current density is closely related to the geometric area of the electrode, where the diffusion layer will theoretically smear out the effect of the roughness of the electrode surface, resulting in the same mass transfer limiting current density of $\sim 6 \text{ mA cm}^{-2}$ at 1600 rpm for catalysts (0.1 M KOH). However, for the catalytic reaction on the actual electrode, many factors, including catalyst selectivity, electrode geometric area, electrolyte viscosity, and oxygen diffusion, could affect the measured limiting current density (*ChemElectroChem.* **7**, 1107 (2020)). During our experiment, we kept our test conditions and geometric area of each catalyst as close as possible. Therefore, considering the catalytic activity of different catalysts in our work, the difference in mass transfer limiting current density is mainly produced by the distinct adsorption and desorption rates of reactive species for various catalysts on the electrode. Furthermore, using *in situ* FTIR and XAFS, we uncover that the coordination-reduced Cl-Co-N₂ active sites in CR-Co/CINC facilitate the adsorption and dissociation of oxygen molecules (O₂) to create the crucial *O intermediate during the ORR process, which leads to a higher mass transfer limiting current density of CR-Co/CINC than that of Pt/C and Co/NC. The NC materials, which lack of metal sites, generally have unsatisfactory ORR catalytic activity in alkaline conditions (*ACS Catal.* **16**, 9366 (2020); *Appl. Catal. B: Environ.* **310**, 121352 (2022)). Thus, there appears to be some differences in the limiting current density among the various samples tested.

Meanwhile, the repeated linear scanning curves of different samples are shown in Fig. N3a. The CR-Co/CINC catalyst displays a better half-wave potential ($E_{1/2}$) than that of the standard Pt/C catalyst (0.932 and 0.857 V vs. RHE, respectively), and a limiting current density of 6.10 mA cm^{-2} . In contrast, the Co/NC and NC exhibit

insufficient ORR electrochemical activity with inferior half-wave potentials (0.787 and 0.732 V vs. RHE, respectively). The above experimental results show that our performance test results are repeatable and reliable.

Fig. N3. (a) Polarization curves for CR-Co/CINC and references under 0.1 M O₂-saturated KOH, 1600 rpm, scan rate of 10 mV s⁻¹. (b) LSV curves at various rotation rates of the CR-Co/CINC catalyst.

Finally, the limiting current of CR-Co/CINC appears to slightly exceed the limiting current, which may be due to background currents, experimental errors in testing and fast reaction kinetics. As shown in Fig. N3b, the limiting current density of the CR-Co/CINC samples is 6.08 ± 0.5 mA cm⁻² after subtracting the background current, slightly higher than the theoretical value of 6.0 mA cm⁻², while it is within 10% of the experimental error (*J. Electrochem. Soc.* **165**, 3001 (2018); *Anal. Chem.* **82**, 6321 (2010)). In addition to the factors such as catalyst loading and electrolyte conditions, the slightly larger limit current density of the CR-Co/CINC may also be attributed to the fast adsorption and desorption rates of reactive species for the CR-Co/CINC catalyst on the electrode.

Accordingly, the original LSV result of CR-Co/CINC has been replaced by a new result after subtracting the background current in the revised manuscript, and Supplementary Fig. 13a in the previous Supplementary Information has been replaced by Fig. N3b as Supplementary Fig. 17a in the revised Supplementary Information. In line 5, page 9 of the revised manuscript, the following text **has been added**: “Furthermore, the mass transfer limiting current density for CR-Co/CINC can reach 6.08 mA cm⁻², which is higher than that observed for the commercial Pt/C catalyst (5.92

mA cm⁻²). This high limiting current density seems to be a consequence of the fast desorption rates of the product on the CR-Co/CINC electrode.”

4. Comment: Based on Fig. S12, the ECSA is very different for all catalysts compared in this study. The authors should try their best to maintain similar level of roughness factor when testing the ORR activity of these materials, as least for CR-Co/CINC and Co/NC.

Reply: We thank the reviewer for the nice question. It is well known that the number of active sites and the intrinsic catalytic sites of the catalyst determine the catalytic activity of the electrocatalyst. The electrochemically active surface area (ECSA) is an important parameter to evaluate the active surface area of an electrochemical reaction (*Int. J. Electrochem. Sci.* **13**, 1173 (2018)), which is substantially different than the geometrical area (A_{Geo}) of the flat electrode (*ACS Catal.* **8**, 6560 (2018)). The ECSA can be obtained by CV curves at different sweep speeds, which are mainly affected by electrode surface roughness, electrode layer thickness and material properties (*Adv. Energy Mater.* **10**, 2000652 (2020)). According to your suggestion, we re-fabricated the electrode to ensure that the surface roughness and the thickness of the electrode are similar, and the obtained C_{dl} and ECSA are shown in Figs. N4, 5 and the relevant results are further analyzed below.

In this work, we used the same electrode with a 3 mm diameter to ensure the same geometric area, and a certain amount of catalyst ink was first sufficiently ultrasonically dispersed, coated and dried on the surface of the polished glassy carbon electrode at room temperature. A uniform sample film is then formed on a glassy carbon electrode to ensure similar electrode surface roughness and thickness for the CS-Co/CINC and Co/NC samples. As shown in Fig. N4a, b, the CR-Co/CINC possesses a C_{dl} value of 15.9 mF cm⁻², close to that of Co/NC (14.8 mF cm⁻²), suggesting similar ECSA values of 111.3 m² g⁻¹ and 103.6 m² g⁻¹ for CR-Co/CINC and Co/NC, respectively. The above results clearly reveal that CR-Co/CINC and Co/NC have similar active specific surface areas, but CR-Co/CINC exhibits better catalytic activity due to its higher intrinsic activity over active sites. This indicates that the source of enhanced activity of CR-

Co/CINC is the Cl–Co–N₄ structure formed by axial Cl coordination, which optimizes the electronic structure to achieve increased intrinsic activity of the active site, rather than the increase in the number of active sites.

Fig. N4. The cyclic voltammograms at different scan rates (5–25 mV/s) and corresponding electrochemical double-layer capacitance (C_{dl}) of the CR-Co/CINC (a), Co/NC (b), Pt/C (c), and NC (d) catalysts.

Fig. N5. ECSA values calculated from potential cycling. The ECSA (m² g⁻¹) can be estimated as the specific value from the gravimetric capacitance C_{dl} by the equation: $ECSA = C / (C_s \cdot L)$, where C_s is the double layer capacitance (F m⁻²) of the glassy carbon electrode surface, for which the typical value of 0.4 F m⁻² was used in KOH solution, and L is the mass of catalyst deposited on the electrode (3.57 m² g⁻¹).

Accordingly, Supplementary Figs. 8–12 in previous Supplementary Information have been replaced by Figs. N4, 5 as Supplementary Figs. 12–16 in the revised Supplementary Information. In line 21, page 9 of the revised manuscript, the following text **has been added**: “*Similar electrochemically active surface area for CR-Co/CINC (111.3 m² g⁻¹) and Co/NC (103.6 m² g⁻¹) indicates that the source of enhanced activity of CR-Co/CINC is the Cl–Co–N₄ structure formed by axial Cl coordination, which optimizes the electronic structure to achieve increased intrinsic activity of the active site, rather than the increase in the number of active sites.*” Furthermore, we have re-optimized the electrode preparation, so the corresponding electrocatalytic performance data were also modified accordingly in the revised manuscript.

5. Comment: Fig. S13: Why is the limiting current of Pt/C never reached?

Reply: Thank you for your nice questions. We really compliment the reviewer for your expert knowledge in the field of electrochemistry. As is known, the commercial Pt/C catalyst is a key reference sample for an efficient four-electron ORR process, and it is necessary to ensure that the Pt/C baseline meets the standard.

Note that the limiting diffusion current density is closely related to the geometric area of the electrode, where the diffusion layer will theoretically smear out the effect of the roughness of the electrode surface, resulting in the same mass transfer limiting current

density of $\sim 6 \text{ mA cm}^{-2}$ at 1600 rpm for catalysts. However, the experimentally measured limiting current density for the $4e^-$ ORR is quite different among various literatures under the same experimental conditions, whether on Pt/C catalysts or non-noble metal catalysts. It is noteworthy that most measured limiting current densities are lower than the theoretical value (*ChemElectroChem.* **7**, 1107 (2020)). Sometimes, the catalyst sample preparation, oxygen intake, impurities, experimental error, and other influencing factors will lead to different current densities (*Electrochimica Acta.* **53**, 3181 (2008); *Anal. Chem.* **82**, 6321 (2010)). Therefore, the thick preparation of the original Pt/C electrode led to a slow mass transfer process on the electrode surface, resulting in its performance not up to standard. According your suggestion, we carefully re-prepare the Pt/C electrode, and repeated linear sweep voltammetry (LSV) for the Pt/C catalyst in the potential region of 0.2–1.1 V vs. RHE is shown in Fig. N6. The well-prepared Pt/C electrode demonstrates a well-developed limiting current density plateau of 5.92 mA cm^{-2} and a superior $E_{1/2}$ of $\sim 0.86 \text{ V}$ under 1600 rpm.

Fig. N6. LSV curves at various rotation rates of Pt/C catalyst.

Accordingly, the original results of Pt/C have been replaced by new results of well-prepared commercial Pt/C in the revised manuscript, and Supplementary Fig. 13b in the previous Supplementary Information has been replaced by Fig. N6 as Supplementary Fig. 17b in the revised Supplementary Information.

6. Comment: Fig. 3: The author should specify the testing conditions in the figure captions including essential information like the type of electrolyte, scan rate, additives, etc.?

Reply: Thank you for your careful inspection and we are sorry for the type of electrolyte, scan rate, additives, etc., was not clearly illustrated in the figure captions. According your suggestion, in line 14, page 8 in the revised manuscript, the text has been added: “Polarization curves for CR-Co/CINC and references under 0.1 M O₂-saturated KOH, 1600 rpm, scan rate of 10 mV s⁻¹.”

7. Comment: Fig. S19: The EXAFS spectra look very different from those presented in the main text and Figure S6, particularly at R values ranging from 2–4 Å

Reply: We thank the reviewer for the insightful suggestion that is quite useful for improving the consistency of this work. To identify the source of the differences in peak intensity at R values ranging from 2–4 Å, we tried several *k* ranges, and it reveals that the peak in the range of 2–4 Å may come from the contribution of a bad data signal-to-noise ratio in the high *k* part of the Co *K*-edge XAFS.

Firstly, it can be found that larger noises appear in the *k* region of 9 and 12.5 Å⁻¹ (Fig. N7a). Subsequently, the three different *k* ranges were selected for the fast Fourier transform, and the results are shown in Fig. N7b. It can be observed that the ~2.6 Å FT peak intensity of the 2.4–10.2 Å⁻¹ region is reduced by 60% in comparison with that of 2.4–11.5 Å⁻¹ region, suggesting an increase of the high-frequency noise in the larger *k* region of 2.4–11.5 Å⁻¹ because of the low Co content for fluorescence XAFS measurement. As the range of *k* decreases to 2.4–9.1 Å⁻¹, the ~2.6 Å FT peak intensity decreases, but a larger value range of *k* is needed to ensure that there are enough independent free points when EXAFS fitting. In the previous manuscript, the *k* range for the fast Fourier transform in EXAFS spectra (Supplementary Fig. 19) was 2.4–10.2 Å⁻¹, but the *k* range of the Fourier transform in the main text and Supplementary Fig. S6 was 2.4–11.5 Å⁻¹. Therefore, the different *k* ranges in the process of the fast Fourier transform lead to differences in EXAFS, especially at R values ranging from 2–4 Å. Finally, to exclude the high-frequency noise in the high *k* region and ensure enough independent free points during EXAFS fitting, the *k* range of 2.4–10.2 Å⁻¹ was selected for all Co samples during the fast Fourier transform. Note that the new fitting results in Fig. N2 show coordination numbers (CNs) of 3.9 and 1.0 for the Co–N bond and an

axial Co–Cl coordination for CR-Co/CINC (see the response to comment 2 for details).

Fig. N7. (a) $k^3\chi(k)$ curves of Co K -edge EXAFS oscillation functions for Co/NC and CR-Co/CINC. (b) Corresponding k^3 -weighted FT of Co K -edge EXAFS oscillation functions at three different k ranges for CR-Co/CINC

Accordingly, Fig. N7 has been added as Supplementary Fig. 7 in the revised Supplementary Information. In line 28, page 5 of the revised manuscript, the following text has been added: “*To exclude the high-frequency noise in the high k region and ensure enough independent free points during EXAFS fitting, the k range of 2.4–10.2 \AA^{-1} was selected for all Co samples during the fast Fourier transform (Supplementary Fig. 7).*”

8. Comment: Fig. 4a: The fittings here again fail to capture the experimental data at R -values in the range of 1.1–1.2 \AA and above 2 \AA . The peaks located above 2 \AA clearly decreased when the potential dropped to 0.9 V while the fits are the same for all three. This might be very important information in longer range coordination. The authors however failed to discuss this part.

Reply: Thank you for your deep consideration. In the previous question, we tried several the several k ranges and it revealed that the peak at the range of 2–4 \AA may come from the contribution of a bad data signal-to-noise ratio in high k part of Co K -edge XAFS. Therefore, the k range of 2.4–10.2 \AA^{-1} was selected for the fast Fourier transform, and the EXAFS data will be reprocessed as shown in Fig. N8. In addition, for single-atom catalysts, we mainly studied the coordination structure information of its first-shell coordination, so the curve fitting was performed on the k^3 -weighted

EXAFS function $\chi(k)$ data in the R -range of 1.0–2.2 Å.

Fig. N8. (a) $k^3\chi(k)$ curves of Co K -edge EXAFS oscillation functions (b) Corresponding k^3 -weighted FT of Co K -edge EXAFS oscillation functions for CR-Co/CINC under different working conditions (ex-situ, 1.00 V, 0.90 V and 0.75 V).

Firstly, to exclude the high-frequency noise in the high k region, the k range of 2.4–10.2 Å⁻¹ was selected for all Co samples during the fast Fourier transform. As shown in Fig. N8b, the peak intensity located above 2 Å shows no significant change when excluding the effects of the high-frequency noise. Subsequently, the EXAFS spectra for the CR-Co/CINC electrocatalyst under different conditions were refitted (Fig. N9 and N10). It exhibits a coordination number of four Co–N bonds and one axial Co–Cl bond under *ex situ* states. Interestingly, when applying a potential of 1.00 V, the coordination number of the Co–N bonds is evidently reduced to two, indicating that the potential-driven structural evolution of Co single sites truly occurs under ORR conditions by releasing Co centers from the NC substrate to form a Cl–Co–N₂ active site. As the potential continually decreases to 0.90 V, the Cl–Co–N₂ coordination combined with an additional Co–O coordination is retained for the Co sites, suggesting the adsorption of key reactive oxygen-containing intermediates. Therefore, the FT curves after transform proceeded in the k range of 2.4–10.2 Å⁻¹ of the CR-Co/CINC electrocatalyst were refitted, successfully avoiding low R differences and large R variations.

Fig. N9. The fitting curve of the k^3 -weighted EXAFS spectrum for CR-Co/CINC under different working conditions (*ex situ*, 1.00 V and 0.90 V).

Fig. N10. (a, c, e, g) The fitting curve of the Co K -edge k^3 -weighted EXAFS spectrum and (b, d, f, h) the $\text{Re}(k^3\chi(k))$ oscillation and fitting curve for CR-Co/CINC under different conditions (*ex situ*, 1.00 V, 0.90 V and 0.75 V).

Accordingly, Fig. 4a has been replaced by Fig. N9 in the revised manuscript and Supplementary Figs. 21–24 in previous Supplementary Information have been replaced by Fig. N10 as Supplementary Figs. 25–28 in the revised Supplementary Information.

In line 28, page 5 of the revised manuscript, the following text **has been added**: “To exclude the high-frequency noise in the high k region and ensure enough independent free points during EXAFS fitting, the k range of 2.4–10.2 \AA^{-1} was selected for all Co samples during the fast Fourier transform.”

9. Comment: Fig. 4a: what exactly is the rational of saying Co-O bonds do not exist for the ex-situ and 1.00 V? The ex-situ XANES seems to be quite similar to CoOOH in Fig. 2c.

Reply: We are greatly grateful to the reviewer for your nice question and careful inspection. To identify the local structure of the Co sites, it can be judged mainly from the oscillation absorption of the XANES results and the position and intensity of the peak appearing in the FT curves (EXAFS data) (*Sci. China Mater.* **58**, 313 (2015)). Firstly, as shown in Fig. N11, the oscillation absorption of the XANES under *ex situ* conditions is close to that under air conditions, suggesting no significant change in the oxidation state. Furthermore, the EXAFS results further demonstrate that there is no obvious change in peak intensity when the CR-Co/CINC electrode is immersed in the electrolyte (*ex situ* conditions), revealing no change in coordination number due to no adsorption of oxygen species. Note that only the shape of the white-line peak of *ex situ* conditions is similar to that of CoOOH, but the oscillation tendency of XANES and the dominant peak of EXAFS are significantly different, which reveals the absence of Co-O coordination for CR-Co/CINC under *ex situ* conditions.

Subsequently, in Fig. N11a, the absorption edge shows a slight positive-energy shift, and the white-line peak displays a slight increase in intensity with an applied voltage of 1.0 V, revealing reduced d -band electron filling. In addition, it can be seen that a dominant peak at ~ 1.56 \AA showed a 20% damping of the peak intensity and a positive shift of 0.04 \AA for CR-Co/CINC as the potential changed from *ex situ* to 1.00 V conditions, which suggests a reduction in the local coordination number. Taking into account the *in situ* infrared results, no adsorption of oxygen-associated species was observed, suggesting that potential-driven reduced coordination of Co single sites occurred at the early reaction state without adsorption of oxygen-containing species.

Above all, oscillation absorption of the XANES results and reduced peak intensity in EXAFS clearly demonstrate a potential-driven reduced coordination of Co single sites and no Co–O coordination exists because of no adsorption by oxygen-containing species at the early reaction state.

Fig. N11. (a) Co *K*-edge XANES spectra and (b) Fourier transforms (FTs) of the Co *K*-edge EXAFS oscillations functions for the CR-Co/CINC under air, *ex situ* and 1.00 V conditions.

Accordingly, in line 14, page 11 in the revised manuscript, the following text **has been added**: “*The reduced peak intensity in EXAFS clearly demonstrates the potential-driven reduced coordination of Co single sites and no adsorption by oxygen-containing species at the early reaction state.*”

10. Comment: Fig. 4b: Is the change presented here reversible?

Reply: We are greatly grateful to the reviewer for your nice question and careful inspection. To determine whether the structural change is reversible, XAFS was performed after the reaction. As shown in Fig. N12a–d, the XANES and EXAFS results before and after the reaction show that the oxidation state and the local coordination structure of Co sites did not change significantly, indicating the stability of the structure. Furthermore, when the applied voltage is removed for a period of time, the XANES and EXAFS results show that the position of the absorption edge and the intensity of the dominant peak in the FT curve could almost return to the initial state (*ex situ* conditions). This result suggests that dynamically-reduced coordination and potential-driven less *d*-band electronic filling during the reaction have returned to their original

state when the ORR potentials are removed. The above results reveal that the potential-driven evolution of electron and coordination structures for the Co active site is a dynamic reversible process.

Fig. N12. (a, c) Co K-edge XANES spectra and (b, d) Fourier transforms (FTs) of the Co K-edge EXAFS oscillations functions for the CR-Co/CINC before and after reaction, *ex situ* and back *ex situ* conditions.

Accordingly, Fig. N12c, d have been added as Supplementary Fig. 31 in the revised Supplementary Information, and in line 28, page 13 of the revised manuscript, the following text **has been added**: “Moreover, *the dynamically-evolved* electron and coordination structures of Co sites return to their original state *after reaction according to the XAFS results (Supplementary Fig. 31), suggesting that the structural evolution of the Co active site is a dynamic reversible process.*”

11. Comment: The authors demonstrate the high performance of the CR-Co/CINC for Zn-air batteries during discharge. However, can this material also survive during charging? The authors should provide such information.

Reply: Thank you for your kind reminder that is very important to improve the integrity

of this work. We conducted galvanostatic discharge measurements under different current densities (Fig. 5d) and the CR-Co/CINC-based ZAB obviously maintains good stability.

To explore whether this material can survive during charging, we performed the galvanostatic charge test for CR-Co/CINC in Fig. N13a. The CR-Co/CINC-based ZAB requires ultralow charge voltages of 1.93, 2.03, and 2.19 V to deliver 2, 5, and 10 mA cm⁻², respectively. In addition, to further evaluate the rechargeability of the Zn-air batteries, we conducted cyclic galvanostatic discharge–charge measurements at a current density of CR-Co/CINC-based ZAB at 10 mA cm⁻² (Fig. N13b). the CR-Co/CINC-based ZAB exhibits an initial discharge potential of 1.31 V and charge potential of 2.19 V with a small charge/discharge voltage gap of 0.88 V. After 48 h of cycling, only a negligible deterioration of the customized ZAB can be recorded, reflecting the durability of the CR-Co/CINC to the corrosive alkaline conditions.

Fig. N13. (a) Charge curves of CR-Co/CINC based ZABs under different current densities. (b) Galvanostatic discharge-charge cycling curve performed under 10 mA cm⁻² for the CR-Co/CINC based ZABs.

Accordingly, Fig. N13 has been added as Supplementary Fig. 34 in the revised Supplementary Information, and in line 11, page 15 of the revised manuscript, the following text **has been added**: “*In addition, the galvanostatic discharge–charge measurements show a negligible deterioration of the discharge–charge voltage, reflecting the promising durability of the CR-Co/CINC in Zn-air batteries (Supplementary Fig. X).*”

12. Comment: The performance of Pt/C in the Zn-air battery is under-rated when

compared with other literature. (For example, *Chemical Engineering Journal*, 2023, 143855). The performance of CR-Co/CINC is therefore not that impressive.

Reply: Thank you for your nice question. In this work, we constructed an atomically dispersed CR-Co/CINC catalyst with symmetry-broken Cl–Co–N₄ moieties via a facile two-step pyrolysis strategy. The well-defined CR-Co/CINC catalyst delivers an appreciable four-electron selectivity and a high kinetic current density exceeding those of Pt/C. Moreover, we have tested the ZAB performance of CR-Co/CINC and Pt/C (for comparison) to investigate their applicability. It can be seen that the CR-Co/CINC-based ZAB presents a higher discharging voltage plateau with a maximum power density close to 176.6 mW cm⁻² (Fig. 5b), which outperforms the Pt/C counterpart (127.1 mW cm⁻²). This indicates the excellent potential of the CR-Co/CINC catalyst for practical ZAB applications.

The power density of catalysts used in zinc-air batteries is related to their electrode distance, oxygen flux, catalyst ink preparation, additives and so on (*J. Energy. Chem.* **44**, 1, (2020)), so it is normal for Pt/C to be in a proper range of power density. As the reviewer noted, the researchers used an air electrode coated with a Pt/C and RuO₂ mixture, in which the PtC+RuO₂ showed a large power density of 208 mW cm⁻² (*Chem. Eng. J.* **469**, 143855 (2023)). For better comparison, we have summarized the maximum power density of the Pt/C and Pt/C mixtures in Table. N1, it can be seen that the power density of commercial Pt/C is not exactly the same but is basically in the range from 100–130 mW cm⁻² (Fig. N14). Meanwhile, we compared the power density of various single-atom catalysts in aqueous Zn-air batteries reported in literatures. As shown in Table N2, the CR-Co/CINC-based ZAB presents a maximum power density of 176.6 mW cm⁻², surpassing most of the reported SACs.

Accordingly, Table N2 has been added in the revised Supplementary Information as Supplementary Table 4. Furthermore, to make our expression more precise, in line 19, page 14 of the revised manuscript, the following text has been added: “*The power density of CR-Co/CINC-based ZAB surpasses most of the reported SACs* (Supplementary Table 4).” And in line 16, page 15 of the revised manuscript, the

following text has been added: “These results suggested that the CR-Co/CINC catalyst has good potential for application in ZABs.”

Table N1. Power density comparison of Pt/C and Pt/C mixture catalysts in aqueous Zn-air batteries reported in literatures.

Catalyst	Power density (mW cm ⁻²)	References
PtC	127.1	This work
PtC ^[1]	110.4	Angew. Chem. Int. Ed. 131 , 5413 (2019)
PtC ^[2]	82	Adv. Funct. Mater. 29 , 1808872 (2019)
PtC ^[3]	141	J. Mater. Chem. A. 9 , 7137 (2021)
PtC ^[4]	106.8	Adv. Mater. 34 , 2105410 (2022)
PtC ^[5]	117.8	Angew. Chem. Int. Ed. 60 , 21237 (2021)
PtC ^[6]	91	Adv. Mater. 32 , 2004670 (2020)
PtC ^[7]	116.3	Energy Storage Mater. 45 , 805 (2022)
PtC ^[8]	128.2	J. Mater. Chem. A. 11 , 6191 (2023)
PtC ^[9]	94.1	ACS Appl. Nano Mater. 6 , 14831 (2023)
PtC ^[10]	120	Small 19 , 2300373 (2023)
PtC ^[11]	73.2	J. Am. Chem. Soc. 142 , 2404 (2020)
PtC ^[12]	144	Adv. Mater. 35 , 2302485 (2023)
PtC ^[13]	115.4	Angew. Chem. Int. Ed. 134 , e202114441 (2022)
PtC ^[14]	124.4	Nano Lett. 21 , 4508 (2021)
PtC ^[15]	136	Adv. Mater. 34 , 2107421 (2022)
PtC ^[16]	104.2	Adv. Mater. 34 , 2202544 (2022)
PtC ^[17]	83.5	Angew. Chem. Int. Ed. 61 , e202117617 (2022)
PtC ^[18]	110.7	Adv. Funct. Mater. 32 , 2203471 (2022)
PtC ^[19]	120	Angew. Chem. Int. Ed. 60 , 27324 (2021)
PtC-RuO ₂ ^[20]	208	Chem. Eng. J. 469 , 143855 (2023)
PtC-RuO ₂ ^[21]	93	J. Mater. Chem. A. 11 , 1894 (2023)
PtC-RuO ₂ ^[22]	110.3	Small Methods 5 , 2000751 (2021)
PtC-IrO ₂ ^[23]	128.8	Angew. Chem. Int. Ed. 132 , 7454 (2020)
PtC-IrC ^[24]	160	Nano Energy 71 , 104597 (2020)
PtC-IrC ^[25]	100.4	Sci. Adv. 8 , 5091 (2022)

Fig N14. Power densities of Pt/C and Pt/C mixture catalysts in aqueous Zn-air batteries reported in literatures, and in the dashed box are Pt/C catalysts with power densities from 100–130 mW cm⁻².

Table N2. Power density comparison of various single-atom catalysts in aqueous Zn-air batteries reported in literatures.

Catalyst	Power density (mW cm ⁻²)	References
CR-Co/CINC	176.6	This work
Co-N ₄ /NC	101.32	Nano-Micro Lett. 13 , 60 (2021)
Fe-AC-2	153	J. Mater. Chem. A. 9 , 7137 (2021)
Fe, Co-SA/CS	86.65	Small Methods 5 , 2000751 (2021)
Fe/OES	186.8	Angew. Chem. Int. Ed. 132 , 7454 (2020)
Co-SAs/@NC	105.3	Angew. Chem. Int. Ed. 131 , 5413 (2019)
Fe-Nx-C	96.4	Adv. Funct. Mater. 29 , 1808872 (2019)
Fe-SAs/NPS-HC	195	Nat. Commun. 9 , 5422 (2018)
Sb SAC	184.6	Angew. Chem. Int. Ed. 60 , 21237 (2021)
Fe/Ni-Nx/OC	148	Adv. Mater. 32 , 2004670. (2020)
FeCo-NSC	152.8	Energy Storage Mater. 45 , 805 (2022)
FeCu-SAC	201.4	J. Mater. Chem. A. 11 , 6191 (2023)
FeNC-SAC-LS	140.18	ACS Appl. Nano Mater. 6 , 14831 (2023)
FeN ₅	159	Small 19 , 2300373 (2023)
Fe-N/P-C-700	133.2	J. Am. Chem. Soc. 142 , 2404 (2020).
Ce SAs/PSNC	212	Adv. Mater. 35 , 2302485 (2023)
Se@NC-1000	176.9	Angew. Chem. Int. Ed. 134 , e202114441 (2022)
SACe-N/PC	155	Nano Lett. 21 , 4508 (2021)
CoNC-SAC	161.8	Sci. Adv. 8 , eabn5091 (2022)
FeCo-DACs/NC	175	Adv. Mater. 34 , 2107421 (2022)
Fe/SNCFS-NH ₃	255.84	Adv. Mater. 34 , 2105410 (2022)
Fe-NiNC	220	Nano Energy 71 , 104597 (2020)
o-MQFe-10: 20: 5	158.2	Angew. Chem. Int. Ed. 61 , e202117617 (2022)
(Zn, Cu)-NC	163.8	Adv. Funct. Mater. 32 , 2203471 (2022)
Fe,Mn/N-C	160.8	Nat. Commun. 12 , 1734 (2021)

Reply to Reviewer #2

In this manuscript, the authors present an atomic coordination-regulated single-atom cobalt catalyst that comprises of asymmetric Co-Cl1N4 moiety, which exhibited excellent electrocatalytic ORR performance compared to the commercial Pt/C catalyst and demonstrated promising application for zinc air battery. The authors did systematic experiments to investigate the structure of as-prepared catalyst and also performed in-situ experiments to study the structural variation during the catalytic reaction. Although the strategy of doping various elements to regulate the coordination of single atom is very common (Adv. Mater. 2023, 35, 2302485; Energy Environ. Sci. 2023, 16, 2629–2636) and the similar materials have been reported in the literatures (J. Am. Chem. Soc. 2022, 144, 14505–14516; Angew. Chem. Int. Ed. 2020, 59, 6122–6127), the mechanism of the enhanced activity discovered by the authors is interesting. Overall, the manuscript is logically organized and the experiments are well designed, however, I feel that the novelty of this work needs to be further clarified, and some other questions also need to be addressed before acceptance.

We are very grateful to Reviewer #2 for highlighting the major findings of this work. After reading your constructive comments and criticisms, we have seriously considered what problems existed in this work and carefully revised the manuscript. It is our great pleasure to receive these important suggestions from the reviewer. We hope the revised manuscript could meet the high standards of *Nature Communications*.

1. Comment: In the Introduction part, the authors describe the design rationale of the single-atom catalysts in detail, i.e., breaking the symmetry of M-N₄ moiety to enhance the four-electron ORR process. As far as I know, there are numerous articles that present the coordination modulation of M-N₄ based single-atom catalysts through heteroatom doping, such as P, O, S, I, Cl doping and so on. Then, what is the new point of this work in terms of catalyst design? And, what are the new findings that can inspire further catalyst design and preparation?

Reply: We thank the reviewer for this constructive suggestion that is quite useful for improving the quality of this work. According to your suggestion, the new point and

new findings in our work will be discussed in detail.

The new point of this work in terms of catalyst design. Firstly, coordination engineering is an efficient approach for improving the performance of single-atom catalysts (SACs), mainly by regulating the direct coordination atoms or environmental atoms. In classical coordination engineering, additional sample sources containing S, P, O, I, or Cl elements are inevitably added in preparation process to first modify the carrier and then regulate the coordination structure of the target metal atom. (*Nat. Commun.*, **11**, 3049 (2020); *Adv. Mater.* **35**, 2302485 (2023); *Energy Environ. Sci.* **16**, 2629 (2023); *Nano Research* **15**, 10063 (2022)). This makes it difficult to accurately regulate the coordination structure of the target element and brings interference from other heterologous elements due to the addition of foreign precursors containing doping elements. Interestingly, in our work, we constructed an atomically dispersed CR-Co/CINC catalyst with symmetry-broken Cl–Co–N₄ moieties via a facile two-step pyrolysis strategy. We used cobalt chloride hexahydrate as our metal precursor, and the metal-anchored NCs were annealed at a low temperature of 300 °C, where the temperature is lower than the complete decomposition temperature of the metal precursor to reserve the ligand, resulting in Cl doping in CR-Co/CINC. In a word, our synthesis process does not introduce other miscellaneous elements, and the key is to directly regulate the coordination structure of the target element. Secondly, unlike the classical planar coordination engineering, axial coordination modification could also endow SACs with novel electronic and chemical properties. A facile two-step pyrolysis strategy was adopted to prepare Co SACs. The morphology and electronic structure show that the axial Cl is riveted at the Co–N₄ sites in the form of a symmetry-broken Cl–Co–N₄ moiety, demonstrating the breaking of the symmetric electron distribution. The axial Cl coordination is more conducive to regulating the electronic filling of the Co *d*-band, realizing a significant enhancement in ORR performance. Finally, *in situ* X-ray absorption spectroscopy reveals that the symmetry-broken Cl–Co–N₄ moiety dynamically evolves into a coordination-reduced Cl–Co–N₂ structure, which effectively optimizes the 3*d* electron filling of Co sites toward a low *d*-band electron

occupancy ($d^{5.8} \rightarrow d^{5.28}$) at the early reaction state, and the dynamic $3d$ electron evolution of the Cl–Co–N₂ structure can promote the cleavage of O–O bond (*OOH) into O* species for a fast four-electron ORR process.

The new findings about catalyst design and preparation. The well-defined classical planar M–N–C configuration of SACs gives them a great chance to catalyze electrochemical reactions. Recently, enormous efforts have been devoted to enhancing the intrinsic activity of SACs by tuning the planar coordination configuration and electronic properties by regulating the planar coordination structure. In this work, as a novel and accessible coordination method, axial coordination engineering can be used to regulate the active site in the axial direction by Cl ligands. Most importantly, by selecting metal precursors (cobaltous chloride) containing coordination elements and controlling the reaction temperature, the axial Cl coordinated over Co sites can be self-regulated without adding additional coordination precursors. Thus, the axial coordination of metal sites can be precisely regulated while avoiding the introduction of miscellaneous elements in the process of synthesis and preparation.

Thus, this work provides a novel and no impurity modification strategy for the design of axial coordination engineering in SACs.

Accordingly, in line 19, page 15 of the revised manuscript, the following text has been added: “*we constructed a kind of atomically dispersed and coordination-regulated CR-Co/CINC catalyst with symmetry-broken Cl–Co–N₄ moieties via a facile two-step pyrolysis strategy, successfully avoiding the introduction of impurities and realizing the coupling of axial Cl over the Co–N₄ moiety.*”

2. Comment: The ORR performance was tested in 0.1 M KOH, which is a strong alkaline electrolyte. As known, the reference electrode used for the alkaline electrocatalytic reaction should be Hg/HgO, but the authors use Ag/AgCl. Indeed, the Ag/AgCl electrode is unstable in the alkaline electrolyte, which may cause inaccurate determination of the reaction potential.

Reply: We thank the reviewer for this constructive suggestion and really compliment the reviewer for your expert knowledge in the field of electrochemistry. ORR

measurements require a stable reference electrode that does not contaminate the experiment and conversely is not contaminated by the experiment. Common reference electrodes include saturated calomel electrodes (SCEs), Hg/HgO electrodes, and Ag/AgCl electrodes.

Considering that the alkaline solution may cause the Ag/AgCl reference electrode to corrode or dissolve under long-term operation, resulting in potential drift or even failure. In our work, we used a double salt bridge Ag/AgCl (saturated KCl) reference electrode, which maximizes the protection of the reference electrode and working electrode in the alkaline electrolyte. Meanwhile, it is worth noting that the Ag/AgCl reference electrode should be calibrated before each use to ensure that its potential does not drift. Furthermore, many works have used Ag/AgCl as a reference electrode in 0.1 M KOH (*Nat. Commun.* **11**, 2831 (2020); *Nat. Commun.* **11**, 3049 (2020); *Energy Environ. Sci.* **12**, 3508 (2019)). This means that the electrochemical experimental results obtained by using a double salt bridge Ag/AgCl reference electrode are also reliable.

According to your suggestion, we have conducted linear sweep voltammetry (LSV) tests for CR-Co/CINC using the Hg/HgO reference electrode in Fig. N15. This demonstrates that the half-wave potential of CR-Co/CINC (Hg/HgO) approaches 0.928 V, with a limiting current density of 6.09 mA cm⁻². This is almost consistent with the results obtained using the Ag/AgCl reference electrode. The above results clearly demonstrate that our electrochemical experimental results are reliable.

Fig. N15. (a) Polarization curves for CR-Co/CINC and Pt/C under 0.1 M O₂-saturated KOH, 1600 rpm, scan rate of 10 mV s⁻¹.

3. Comment: Can the authors determine the weight ratio of Cl in the CR-Co/CINC catalyst? And, the authors showed C-Cl coordination in Cl 2p XPS, this is not consistent with the structure model used in Figure 1e demonstrating Co-Cl coordination. In addition, in the XPS survey spectrum of Co/NC catalyst that is supposed to be free of Cl atoms, however, a weak Cl 1s peak also appeared (Figure S4a). Is there an explanation for this?

Reply: Thank you for your nice questions, and we are sorry for that the weight ratio of Cl was not verified in our manuscript. The weight ratio of Cl in the CR-Co/CINC catalyst was determined to be 2.15 wt % based on X-ray photoelectron spectroscopy (XPS) data.

In this work, we constructed an atomically dispersed CR-Co/CINC catalyst with symmetry-broken Cl–Co–N₄ moieties via a facile two-step pyrolysis strategy. We used cobalt chloride hexahydrate as our metal precursor, and the metal-anchored NCs were annealed at a low temperature of 300 °C, where the temperature is lower than the complete decomposition temperature of the metal precursor to reserve the ligand, resulting in Cl doping in CR-Co/CINC. A portion of Cl would inevitably bond to C in the NC substrate under the pyrolysis process. This can also be demonstrated by the high-resolution Cl 2p XPS spectrum (Supplementary Fig. 4b) (*Energy Environ. Sci.* **11**, 2348, (2018); *Nat. Commun.* **10**, 2980 (2019)). Furthermore, the Co/NC was prepared by pyrolysis at 800 °C, the ligand Cl in the metal precursor was removed, and the weak Cl peak (Supplementary Fig. 4a) that appeared may be attributed to the partial remaining C–Cl coordination. Above all, the enhanced ORR activity is mainly due to the optimization of the electronic structure of the Co site after introduction of an axial Cl coordination for CR-Co/CINC and is independent of the C–Cl coordination in the carrier. Therefore, the structure model used in Fig. 1e is to highlight the local structure of the Co metal site and axial Cl coordination. Therefore, the local structure model of Co in Fig. 1e does not conflict with the existence of the C–Cl bond. Meanwhile, a weak Cl 1s peak (Supplementary Fig. 4a) can be attributed to a small amount of C–Cl coordination in Co/NC. The Co/NC was prepared by pyrolysis at 800 °C, in which the

ligand Cl in the metal precursor was removed, but the partial remaining C–Cl coordination was due to strong C–Cl coordination.

Accordingly, in line 22, page 5 of the revised manuscript, the following text **has been added**: “*The content of Cl in the CR-Co/CINC is assessed as 2.15 wt % by XPS.*” In line 28, page 5 of the revised manuscript, the following text **has been added**: “*To highlight the local structure of Co metal site and axial Cl coordination, the local structure model is shown in Fig. 1e.*”

4. Comment: For Co K-edge, the authors should compare the adsorption edge to confirm the valence state (Nat. Commun. 2022, 13, 7754; ACS Catal. 2022, 12, 3138–3148). As depicted in Figure 2c, the adsorption edges of Co/NC and CR-Co/CINC are located between Co foil (0) and CoOOH (+3). However, in Figure S7, their oxidation states are calculated to be larger than 3.

Reply: We thank the reviewer for the insightful suggestion that is quite useful for improving the consistency of this work. The oxidation state of the transition metal can be obtained by linear fitting of the position of the absorption edge of the metal (Nat. Commun. **11**, 1029 (2020)). The position of the metal absorption edge can be determined by the maximum value of the first derivative of XANES. As shown in Fig. N16a, the high-energy shift of the absorption edge and enhanced intensity of the white-line peak in CR-Co/CINC demonstrate an increased valence state of Co after Cl introduction. Furthermore, as shown in the blue box in the Co K-edge XANES, a slightly higher absorption edge position of CR-Co/CINC was observed compared to CoOOH. In order to further clearly obtain the position of the absorption edge, the first derivative curve is transformed in Fig. N16b. The maximum value of the first derivative of CR-Co/CINC is larger than that of CoOOH, revealing a higher Co oxidation state in CR-Co/CINC related to CoOOH. Above all, the adsorption edges of Co/NC and CR-Co/CINC are higher than that of CoOOH based on the maximum value of the first derivative, suggesting a higher oxidation state.

Fig. N16. (a) Co K-edge XANES spectra and (b) Normalized derivative curves for CR-Co/CINC and reference samples.

Accordingly, Fig. N16 has been added as Supplementary Fig. 10 in the revised Supplementary Information, and in line 27, page 7 of the revised manuscript, the following text **has been added**: “*The maximum value of the first derivative of CR-Co/CINC is larger than that of CoOOH, revealing a higher Co oxidation state in CR-Co/CINC related to CoOOH (Supplementary Fig. 10).*”

5.1 Comment: Some experimental details need to be added. For Figure 3f, the author should add how to conduct the CA test. Whether there is the oxygen continuously purged into the solution or not during the stability test? Indeed, it seems more common to use the CV scanning method to conduct the ORR stability test.

Reply: Thank you for your insightful comment and constructive suggestion that is quite useful for improving the quality of this work. For the current-time chronoamperometric test, O₂ was bubbled into 0.1 M KOH electrolyte for 30 min prior to the experiment and a flow of O₂ was maintained over the electrolyte during the test to ensure O₂ saturation. The test process was constant at 0.7 V vs RHE. In addition, we have added a CV test to determine the stability of CR-Co/CINC. Seen from Fig. N17, the CR-Co/CINC demonstrates significant stability with only a loss of 20 mV in E_{1/2} after 10000 cycles.

Fig. N17. ORR polarization curves of CR-Co/CINC before and after 10000 potential cycles. Inset is the cyclic voltammetry (CV) data for CR-Co/CINC before and after 10000 potential cycles under 50 mV s^{-1} .

According to your suggestion, in line 22, page 17 of the revised manuscript, the following text has been added: “*For the current-time chronoamperometric test, O_2 was bubbled into 0.1 M KOH electrolyte for 30 min prior to the experiment and a flow of O_2 was maintained over the electrolyte during the test to ensure oxygen saturation. The test process was constant at 0.7 V vs RHE.*” In addition, Supplementary Fig. 17 in previous Supplementary Information has been replaced by Fig. N17 in the revised Supplementary Information as Supplementary Fig. 21, and in line xx, page xx of the revised manuscript, the following text has been added: “*Additionally, the stability and durability of the CR-Co/CINC catalyst were measured via chronoamperometry (CA) and accelerated durability testing (ADT). The CR-Co/CINC exhibits reliable stability with better methanol resistance and lower current density attenuation (<8%) after 120 h of CA testing at 0.7 V, and the CR-Co/CINC demonstrates only a loss of 20 mV in $E_{1/2}$ after the ADT, suggesting its long-term durability for the ORR (Fig. 3f and Supplementary Figs. 20, 21).*”

5.2 Comment. Furthermore, I am confused about the characterizations in Figure S19, shown that there is almost no change happened on the catalyst after ORR stability test. However, the decreased valences states of CR-Co/CINC with the decreased potentials can be obviously seen in Figure S25.

Reply: We are greatly grateful to the reviewer for your nice question and careful inspection. The potential-driven evolution of electron and coordination structures for

the Co active site is a dynamic reversible process. To determine whether the structural change is reversible, the XAFS was performed after the reaction. As shown in Fig. N18 a and b, the XANES and EXAFS results before and after the reaction show that the oxidation state and the local coordination structure of Co sites did not change significantly, indicating the stability of the structure. Furthermore, when the applied voltage is removed for a period of time, the XANES and EXAFS results in Figs. N18c and d show that the position of the absorption edge and the intensity of the dominant peak in the FT curve return to the initial state (*ex situ* conditions). This result suggests that dynamically-reduced coordination and potential-driven less *d*-band electronic filling during the reaction have returned to their original state when the ORR potentials are removed. The above results reveal that the potential-driven evolution of electron and coordination structures for the Co active site is a dynamic reversible process. The results of the dynamic evolution of structures along with reversible structural changes resulting from voltage drives have been reported (*Nat. Commun.* **6**, 6118 (2021)).

Fig. N18. (a, c) Co *K*-edge XANES spectra and (b, d) Fourier transforms (FTs) of the Co *K*-edge EXAFS oscillations functions for the CR-Co/CINC before and after reaction, *ex situ* and back *ex situ* conditions.

Accordingly, in line 28, page 13 of the revised manuscript, the following text **has been added**: “Moreover, *the dynamically-evolved* electron and coordination structures of Co sites return to their original state *after reaction according to the XAFS results (Supplementary Fig. 31), suggesting that the structural evolution of the Co active site is a dynamic reversible process.*”

5.3 Comment. Also, a mistake was made in Figure S25, in which the figures are not corresponded by the figure captions. Additionally, the electrochemical method used for conducting the in-situ XAS test should also be added.

Reply: Thank you for your careful inspection. According to your suggestion, we have corresponded to the figure captions in Supplementary Fig. 25. In addition, we checked the full manuscript carefully to make sure that there were no similar mistakes.

The electrochemical method used for conducting the *in situ* XAS test was included in the manuscript. In line 12, page 19 in the revised manuscript, the following text **has been added**: “*In situ XAFS measurements. The Co K-edge XAFS data were collected at the IWIB station in the Beijing Synchrotron Radiation Facility (BSRF), China. The storage ring of BSRF was operated at 2.5 GeV with a maximum current of 250 mA. The beam from the bending magnet was monochromatized utilizing a Si (111) double-crystal monochromator and further detuning of 30% to remove higher harmonics. In situ XAFS measurements were performed with Co catalyst-coated carbon cloths in alkaline solution by a smart homemade cell through a three-electrode system. In particular, the 5 mg sample was uniformly dispersed in a 1 ml solution (0.5 mL ethanol and 0.5 mL deionized water) with 20 μ L Nafion. Then, the catalyst ink was dropped onto the carbon paper as the working electrode ($\sim 1\text{cm} \times 1\text{cm}$), and the back of the carbon paper was fixed with Capton film to ensure that all electrocatalysts could react with the electrolyte. To obtain the dynamic evolution information of the active site during the electrochemical reaction, a series of representative voltages (1.00–0.75 V) were applied to the electrode. During the collection of XAFS measurements, the position of the absorption edge (E_0) was calibrated using a standard sample of Co foil, and all XAFS data were collected during one period of beam time. The XAFS spectra were collected*”

through a Lytle detector (fluorescence mode) to obtain weak signals in the electrochemical reaction process.”

6. Comment: The authors clarified that the introduction of Cl can in situ optimize the 3d electron filling of Co sites towards a low d-band electron occupancy (d5.8→d5.28), which promotes the formation of O* and enhances 4e- ORR selectivity. To make this conclusion more convincing, I suggest the d-band electron occupancy of Co/NC with symmetric Co-N4 moiety should be calculated.

Reply: We thank the reviewer for this constructive suggestion that is quite useful for improving the quality of this work. To make our conclusion more convincing, we conduct *in situ* XAFS measurements of Co/NC and calculate the corresponding *d*-band electron occupancy in Fig. N19. *In situ* XANES spectra of the Co *K*-edge for Co/NC show that the absorption edge shows a very slight positive-energy shift, and the white-line peak displays a slight increase in intensity with an applied voltage of 1.0 V, compared with the *ex situ* conditions. To quantify the change in *d*-band electron filling, we correlate the absorption edge shifts with the *d*-band electron counts of Co, as shown in Fig. N19b. The Co 3*d* electron filling count in Co/NC (5.96) decreases as the operating state changes from *ex situ* (5.96) to 1.00 V (5.91) conditions. However, the change in the Co 3*d* electron filling count in Co/NC (0.05) is significantly lower than that of CR-Co/CINC (0.52 electrons), which is beneficial for optimizing the adsorption kinetics of intermediates for CR-Co/CINC because of more vacant *d*-band orbitals for coupling with O 2*p* orbitals. Above all, the Co 3*d* electron filling count in CR-Co/CINC depopulates much more rapidly and violently with the applied potential than that of Co/NC, suggesting a faster reaction kinetics.

Fig. N19. (a) XANES spectra of the Co *K*-edge recorded at different applied potentials during the ORR process for Co/NC. Inset, magnified absorption edge region. (b) The fitted average formal *d*-band electron counts of Co at Co/NC under *ex situ*, 1.00 V, and 0.90 V conditions based on the absorption edge of Co *K*-edge XANES spectra.

Accordingly, Fig. N19 has been added as Supplementary Fig. 30 in the revised Supplementary Information, and in line 25, page 12 of the revised manuscript, the following text has been added: “Notably, the Co 3*d* electron filling count in CR-Co/CINC depopulates much more rapidly and violently with the applied potential than that of Co/NC (Supplementary Fig. 30), which is beneficial for optimizing the adsorption kinetics of intermediates.”

7. Comment: The phenomena found by the in-situ XAFS and SRIR are interesting, the signals of O* and OOH* are obvious in Figures 4d and 4e. Indeed, in the recent literature (J. Am. Chem. Soc. 2022, 144, 14505–14516), they clarified that the pyrrole-type Co-N₄ is mainly responsible for 2e⁻ ORR to generate H₂O₂, while the pyridine-type Co-N₄ can effectively catalyze the 4e⁻ ORR. Therefore, I am wondering if there is any possibility that the Co/NC synthesized in this work is pyrrole-type and the incorporation of Cl will change its nature to the pyridine-type? I recommend the authors should investigate the detailed structures of the supports.

Reply: Thank you for your careful inspection and it is useful for improving the clarity of this work. The researchers used the nitrogen precursor of 4-dimethylaminopyridine to construct pyrrole-type Co-N₄ and 2-methylimidazole to obtain pyridine-type Co-N₄ (J. Am. Chem. Soc. **144**, 14505 (2022)). In our work, we used 2-methylimidazole as our

N precursor, and the relative ratios of the deconvoluted N species for CR-Co/CINC and Co/NC calculated from the high-resolution N 1s XPS spectra are summarized in Fig. N18. The CR-Co/CINC delivers a pyrrole-type N content of 22.9 %, close to that of Co/NC (24.2%), and the proportions of pyridinic-N and Co-N are also very close. The small difference would not affect its ORR selectivity. Moreover, the Co-N peak of CR-Co/CINC displays a slight positive shift of ~ 0.2 eV in comparison with Co/NC (Fig. 2b), suggesting the reduced interaction between Co and N attributed to the introduced Cl atoms. Using *in situ* XAFS and SRIR measurements, the symmetry-broken CoCl_1N_4 moiety in CR-Co/CINC favors the cleavage of the O–O bond ($^*\text{OOH}$) into O^* species for a fast four-electron ORR process, delivering an outstanding ORR performance with a large half-potential of 0.93 V versus RHE. That is, the excellent ORR performance mainly comes from the symmetry-broken Cl-Co-N₄ moiety and rapidly optimized 3d electron filling of Co sites after inducing Cl.

Fig. N20. Proportions of different N species in the CR-Co/CINC and Co/NC catalysts.

Accordingly, Fig. N20 has been added in the Supplementary Information as Supplementary Fig. 9, and in line 13, page 7 of the revised manuscript, the following text has been added: “*The corresponding ratios of the deconvoluted N species are summarized in Supplementary Fig. 9. It shows the types of N with similar proportions, excluding the influence of the N type in the substrate on ORR performance.*”

8.1 Comment: The details of ZAB test should also be added, such as the flow rate of the O₂-saturated electrolyte, the effective area of the Zn plate and the working electrode.

Reply: Thank you for your nice comment. It is useful for improving the quality of this work. In this work, the liquid ZAB was tested in a cyclically home-made instrument under an ambient atmosphere. The electrolyte was composed of 6 M KOH with 0.2 M zinc acetate, and a flow of O₂ was maintained (20 sccm) into the electrolyte during the test to ensure O₂ saturation. The catalysts coated on the carbon paper were used as the membrane electrode assembly (MEA) of the cathode (catalysts coating area was controlled at 1 cm × 1 cm), and a zinc plate with an effective area of 1 cm × 1 cm was used as the anode.

Accordingly, in line 28, page 18 of the revised manuscript, the details of ZAB test have been added: *“The liquid ZAB was tested in a cyclically home-made instrument under an ambient atmosphere. The electrolyte was composed of 6 M KOH with 0.2 M zinc acetate, and a flow of O₂ was maintained (20 sccm) into the electrolyte during the test to ensure O₂ saturation. The catalysts coated on the carbon paper were used as the membrane electrode assembly (MEA) of the cathode (catalysts coating area was controlled at 1 cm × 1 cm), and a zinc plate with an effective area of 1 cm × 1 cm was used as the anode”*

8.2 Comment. Actually, the discharge curve was carried out for only 10 min under each current density, so it is not appropriate to describe the CR-Co/CINC possesses a high stability in ZAB. In addition, the figure caption is not correctly describing the Figure S26.

Reply: Thank you for your nice questions and careful inspection. Galvanostatic discharge observations at current densities ranging from 5 to 50 mA cm⁻² are shown in Fig. 5d. CR-Co/CINC-based ZABs obviously maintain higher discharge voltages than Pt/C-based ZABs under the current density ranges and then periodically return to 5 mA cm⁻². Following a high-rate discharge at 50 mA cm⁻², the discharge potential for the CR-Co/CINC-based ZAB recovers to the starting stage (5 mA cm⁻²), indicating that the CR-Co/CINC-based ZAB possesses a good rate capability and good reversibility, which can be attributed to the efficient ORR activity and good stability of the CR-Co/CINC catalyst. These results indicate the excellent potential of the CR-Co/CINC catalyst for

practical ZAB applications.

Furthermore, the discharging curve with three recharging cycles under 10 mA cm^{-2} of CR-Co/CINC-based ZABs is plotted in Fig. N21. This CR-Co/CINC-based ZAB can work stably through mechanical recharging and requires only replenishment of the consumed zinc anode and electrolytes. No apparent decline in the output voltage is observed after three charging cycles. These results reveal that a unique CR-Co/CINC catalyst that maintains satisfactory catalytic activity in practical operation has good application prospects in industrial ORR. Additionally, we thank the reviewer again for pointing out the incorrectly description of Supplementary Fig. 26, and a proper description has been added in the figure caption.

Fig. N21. Discharging curve with three recharging cycles of CR-Co/CINC under 10 mA cm^{-2} .

Accordingly, Fig. N21 has been added as Supplementary Fig. 33 in the revised Supplementary Information, and in line 11, page 15 of the revised manuscript, the following text has been added: “*Noticeably, no discernible degradation was recorded after 3 cycles for ZAB with the CR-Co/CINC electrocatalyst over a duration of 30 h at 10 mA cm^{-2} (Supplementary Fig. 33).*” In addition, the figure caption describing Supplementary Fig. 32 in the revised Supplementary Information has been added: “(a) *Digital photograph of the Zn-air battery constructed using the CR-Co/CINC catalyst as the cathode catalyst. (b) Top view and detailed description of the homemade ZAB.*”

REVIEWER COMMENTS

Reviewer #1 (Remarks to the Author):

In general, the author has addressed most of the comments very seriously and thoroughly. The quality of the revision is significantly improved compared to the original manuscript. Comprehensive materials characterization and the electrochemical measurements were conducted to render a strong and solid work. The revision is suitable for publication but would be even better if the authors can address the following comments.

1. When addressing comment 1, the author stated the difficulty in in-situ EXAFS measurements. However, Fig. 1e doesn't seem to be acquired under in-situ conditions. It is hard to understand why the author is stressing this point here.
2. In Fig. N3, the onset potentials for Pt/C and CR-Co/CINC are almost identical, which implies similar kinetics and the difference in half-wave potential is likely due to the non-idealness of the deposited electrode. Is this data iR-corrected? If so, the authors should also provide the uncorrected data and the measured impedance in the supporting information.
3. The ECSA of Co/NC has doubled in the revision but the pol-curve remains almost the same. This is quite unconventional.
4. The authors apparently misunderstood the reviewer's comment on the roughness factor. When talking about roughness factor, one means that the total surface area (not the ECSA) of the working electrodes should be kept close for fair comparison. The ECSA is a materials property which should be constant for the same material. It is very weird to see that the ECSA of Co/NC has doubled in the revision.

Reviewer #2 (Remarks to the Author):

The authors addressed well my comments. I would recommend its acceptance.

Response to Reviewers' Comments

We are grateful to the reviewers for having given us important and valuable comments on the manuscript NCOMMS-23-41656A entitled: “*In situ* modulated coordination fields of single-atom cobalt at the early reaction state for enhanced oxygen reduction reaction”. The detailed replies to your comments are presented in a point-to-point manner as follows. **The modifications in the revised manuscript are highlighted in the yellow background.**

Reviewer #1 (Remarks to the Author):

In general, the author has addressed most of the comments very seriously and thoroughly. The quality of the revision is significantly improved compared to the original manuscript. Comprehensive materials characterization and the electrochemical measurements were conducted to render a strong and solid work. The revision is suitable for publication but would be even better if the authors can address the following comments.

Comment 1. When addressing comment 1, the author stated the difficulty in in-situ EXAFS measurements. However, Fig. 1e doesn't seem to be acquired under in-situ conditions. It is hard to understand why the author is stressing this point here.

Reply: Thank you for your nice question and careful inspection. We are sorry that the difficulty in *in-situ* EXAFS measurements was stated in the previous response letter. My original intention was to express that the XAFS measurements of single atom catalyst in the air state and working state has bad data signal-to-noise ratio in the high k part of Co K -edge XAFS. Similarly, in XAFS measurements of static samples, the low content of metal elements also causes bad data signal-to-noise ratio in the high k part of Co K -edge XAFS. Therefore, the peaks in the range of 2–4 Å are indeed from the bad data signal-to-noise ratio in the high k part during the fast Fourier transform, and the refitting curves match well in the range of 1–2 Å in the Co K -edge EXAFS.

Comment 2. In Fig. N3, the onset potentials for Pt/C and CR-Co/CINC are almost

identical, which implies similar kinetics and the difference in half-wave potential is likely due to the non-idealness of the deposited electrode. Is this data iR -corrected? If so, the authors should also provide the uncorrected data and the measured impedance in the supporting information.

Reply: We thank the reviewer for the nice question. Firstly, all LSV curves results were obtained without iR -correction during ORR measurements. As is known, the onset potential is an important parameter to evaluate the intrinsic activity of catalysis (*Electrochem. Commun.* **38**, 142 (2014)), which can be used to assess the activation and adsorption capacity of the reactants prior to the actual catalytic reaction. The onset potential can be defined as the potential at which the ORR current is 5% of the diffusion-limited current during ORR process (*Angew. Chem. Int. Ed.* **55**, 2650 (2016); *Adv Energy Mater.* **4**, 1301523(2014)). In this work, the onset potential of 1.008 V was required for CR-Co/CINC to achieve a current density of 0.304 mA cm^{-2} (Figure N1), slightly outperforming that of Pt/C (0.994 V at 0.296 mA cm^{-2}). Furthermore, a smaller Tafel slope of CR-Co/CINC related to Pt/C further indicates the faster reaction kinetics because of breaking the symmetric electron distribution for the CR-Co/CINC. Therefore, the slightly higher onset potential and the faster reaction kinetics of CR-Co/CINC quickly aroused the ORR reaction with a larger half-wave potential compared to Pt/C.

Figure N1. (a) Polarization curves for CR-Co/CINC and Pt/C under 0.1 M O₂-saturated KOH, 1600 rpm. (b) The onset potential can be determined based on the LSV curve when the ORR current is 5% of the diffusion-limited current.

Accordingly, Figure N1 has been added in the revised Supplementary Information as

Supplementary Fig. 12. In line 5, page 9 of the revised manuscript, the following text has been modified: “*The onset potential of 1.008 V was required for CR-Co/CINC to achieve 5% of the diffusion-limited current (Supplementary Fig. 12), slightly outperforming that of Pt/C (0.994 V).*”

Comment 3. The ECSA of Co/NC has doubled in the revision but the pol-curve remains almost the same. The is quite unconventional.

Reply: We thank the reviewer for this constructive suggestion and really compliment the reviewer for your expert knowledge in the field of electrochemistry. The ECSA of samples by C_{dl} would be affected by many factors, including the loading of catalyst, the uniformity of electrode film, the flatness of the surface of glassy carbon electrode, and the penetration of electrolyte on the electrode surface. But, in the ECSA test of Co/NC of the first version, we used fresh electrodes to directly conduct CV tests at different sweep speeds to obtain C_{dl} . The influence of electrode surface penetration was not taken into account in the test, so the ECSA test result of Co/NC was too small. However, the polarization curve test was obtained after multiple cycles of CV activation and the electrode surface was fully impregnated by the solution, so the polarization curve results were accurate in the first version. We apologize for the inaccuracy of the electrochemical test at the beginning of our work. Then, inspired by the reviewer's guidance, the catalyst was re-tested. Specifically, the electrode activation was performed prior to the polarization curve and ECSA test to ensure adequate solution infiltration. As a result, ECSA was increased in the revision, and the polarization curve is close to the original version, because the testing conditions are guaranteed to be the same this time and close to the test state of the last polarization curve.

Comment 4. The authors apparently misunderstood the reviewer's comment on the roughness factor. When talking about roughness factor, one means that the total surface area (not the ECSA) of the working electrodes should be kept close for fair comparison. The ECSA is a materials property which should be constant for the same material. It is

very weird to see that the ECSA of Co/NC has doubled in the revision.

Reply: We thank the reviewer for this constructive suggestion and really compliment the reviewer for your expert knowledge in the field of electrochemistry. After the reviewer's guidance and careful inquiry of ECSA related literature, we found that the ECSA of samples by C_{dl} would be affected by many factors, including the loading of catalyst, the uniformity of electrode film, the flatness of the surface of glassy carbon electrode, and the penetration of electrolyte on the electrode surface. In order to further ensure the reliability of the ECSA test results, we selected the same glass carbon electrode to support the CR-Co/CINC and Co/NC with the same catalyst load. Before the test, 30 cycles of CV tests were required to ensure that the sample on the electrode surface was fully infiltrated. Each sample is tested the three times in the repetitive process to obtain reliable C_{dl} . As shown in Figures N2 and N3, the CR-Co/CINC possesses C_{dl} values of 15.04, 16.89 and 16.21 $mF\ cm^{-2}$. For comparison, the Co/NC delivers C_{dl} values of 14.88, 13.39 and 14.56 $mF\ cm^{-2}$. In Figure N4, the ECSA average value of CR-Co/CINC ($112.4\ m^2\ g^{-1}$ with an error bar of $7.01\ mF\ mg^{-1}$) is slightly larger than Co/NC ($100.1\ m^2\ g^{-1}$ with an error bar of $6.21\ m^2\ g^{-1}$), which is also closer to the results of the revision. The similar ECSA is mainly due to the similar BET and metal loading of the two materials. In this process, we tried to maintain the consistency of electrode surface roughness, sample load, electrode film uniformity and electrode penetration to eliminate the influence of test errors on ECSA results. Above all, the results of ECSA obtained so far in three replicates should be reliable.

Figure N2. The cyclic voltammetry curves of CR-Co/CINC catalyst at different scan rates (5–25 mV/s) (a) first, (c) second, (e) third and corresponding electrochemical double-layer capacitance (C_{dl}) (b) first, (d) second, (f) third.

Figure N3. The cyclic voltammetry curves of Co/NC catalyst at different scan rates (5–25 mV/s)

(a) first, (c) second, (e) third and corresponding electrochemical double-layer capacitance (C_{dl}) (b) first, (d) second, (f) third.

Figure N4. (a) C_{dl} results from three repeated tests for CR-Co/CINC and Co/NC. (b) ECSA values calculated from C_{dl} . The error bar is obtained by three repeated tests.

Accordingly, the Supplementary Figs. 12 and 13 have been replaced by Figures N2 and N3 as Supplementary Figs. 13 and 14, and Figure N4 has been added in the revised Supplementary Information as Supplementary Fig. 15. In reverse line 8, page 9 of the revised manuscript, the following text has been modified: “*To eliminate the influence of test errors, including the loading of catalyst, the uniformity of electrode film, the flatness of the surface of glassy carbon electrode, and the penetration of electrolyte on the electrode surface, CR-Co/CINC and Co/NC are tested three times in the repetitive process for obtaining the error bars of ECSA (Supplementary Figs. 13–15). Moreover, similar electrochemically active surface area (ECSA) for CR-Co/CINC ($112.4\ m^2\ g^{-1}$ with an error bar of $7.01\ mF\ mg^{-1}$) and Co/NC ($100.1\ m^2\ g^{-1}$ with an error bar of $6.21\ m^2\ g^{-1}$) indicates that the source of enhanced activity of CR-Co/CINC is the Cl–Co– N_4 structure formed by axial Cl coordination.*”

Reviewer #2 (Remarks to the Author):

The authors addressed well my comments. I would recommend its acceptance.

Reply: Thank you very much for your approval and guidance on our manuscript.